



# Accuracy Assessment of MODIS Land Aerosol Optical Thickness Algorithms using AERONET Measurements

Hiren Jethva[1,2], Omar Torres[2], Yasuko Yoshida[3]

[1]Universities Space Research Association, 7178 Columbia Gateway Drive, Columbia, MD 21046, USA

[2]NASA Goddard Space Flight Center, Earth Science Division, Code 614, Greenbelt, MD 20771, USA

[3]Science Systems and Applications, Inc., 10210 Greenbelt Rd, Lanham, MD 20706 USA

**Abstract**

The planned simultaneous availability of visible and near-IR observations from the geostationary platforms of Tropospheric Emissions: Monitoring of Pollution (TEMPO) and GOES R/S Advanced Base Imager (ABI) will present the opportunity of deriving an accurate aerosol product taking advantage of both ABI's high spatial resolution in the visible and TEMPO's sensitivity to aerosol absorption in the near-UV. Because ABI's spectral coverage is similar to that of MODIS, currently used MODIS aerosol algorithms can be applied to ABI observations. In this work, we evaluate existing MODIS algorithms of that derive aerosol optical thickness (AOT) over land surfaces using visible and near-IR observations. The Dark Target (DT), Deep Blue (DB), and Multiangle Implementation of Atmospheric Correction (MAIAC) algorithms are all applied to Aqua-MODIS radiance measurements. We have carried out an independent evaluation of each algorithm by comparing the retrieved AOT to space-time collocated ground-based sunphotometer measurements of the same parameter at 171 sites of the Aerosol Robotic Network (AERONET) over North America (NA). A spatiotemporal scheme co-locating the satellite retrievals with the ground-based measurements was applied consistently to all three retrieval datasets. We find that while the statistical performance of all three algorithms is comparable over darker surfaces over eastern NA, the MAIAC algorithm provides relatively better comparison over western NA sites characterized by inhomogeneous elevation and bright surfaces. MAIAC's finer product resolution (1 km), allows a substantially larger number of matchups than DB 10-km and DT 10-km (DT 3-km) products by 108% and 125% (83%) respectively over Eastern NA, and by 144% and 220% (195%) over Western NA. The



characterization of error in AOT for the three aerosol products as a function of MAIAC-retrieved bi-directional surface reflectance shows a systematic positive bias in DT retrievals over brighter surfaces, whereas DB and MAIAC retrievals showed no such bias throughout the wide range of surface brightness with MAIAC offering lowest spread in errors. The results reported here

5 represent an objective, unbiased evaluation of existing over-land aerosol retrieval algorithms of MODIS. The detailed statistical evaluation of the performance of each of these three algorithms may be used as guidance in the development of inversion schemes to derive aerosol properties from ABI or other MODIS-like sensors.

**Keywords:** aerosol optical thickness, MODIS, Dark Target, Deep Blue, MAIAC, AERONET,

10 North America



# 1. Introduction

The Tropospheric Emissions: Monitoring of Pollution (TEMPO) mission is NASA's first Earth Venture Instrument (*Zoogman et al.,* 2017). It will be hosted on a still undetermined geostationary satellite with an estimated earliest launch in 2020. TEMPO's hyperspectral observations in the 290-490 nm and 540-740 nm wavelength ranges (0.6 nm spectral resolution) will measure trace gas concentrations ($O_3$, $NO_2$, $SO_2$, $CH_2O$, and others) and suspended particle matter (PM). Spatial coverage includes most of Canada, the Contiguous United States (CONUS), Northern Mexico, and part of the Caribbean at an approximate spatial resolution of 2.1x4.7 km$^2$. TEMPO partially fulfills the objectives of the Geostationary Coastal and Air Pollution Events (GEO-CAPE) mission recommended by National Research Council's Earth Science Decadal Survey to measure tropospheric gases, aerosols, and coastal phytoplankton to monitor air and water quality (*Fishman et al*., 2012).

Accurate characterization of the tropospheric aerosol load is required as input to a PM computational scheme along with meteorological information such as temperature and pressure profiles, relative humidity, and planetary boundary layer (PBL) height. The simultaneous availability on GEO platforms of TEMPO and GOES R/S Advanced Baseline Imager (ABI) observations in the visible and near-IR present the opportunity of deriving an accurate aerosol product taking advantage of both ABI's high spatial resolution in the visible and near-IR, and TEMPO's sensitivity to aerosol absorption in the near-UV. The combination of sub-kilometer spatial resolution and the multi-spectral observational capability make the ABI an optimum sensor for the derivation of an aerosol optical thickness (AOT) product over land at the GEO-CAPE required accuracy (*Fishman et al*., 2012) to be used in conjunction with TEMPO observations for air quality and climate applications.

Satellite-based aerosol remote sensing has been an essential tool to monitor the spatial and temporal distributions of aerosols over the globe. Significant advancements in aerosol retrieval capabilities over both land and oceans have taken place over the last 20 years. The deployment of the Moderate Resolution Imaging Spectrometer (MODIS) and the Multi-angle Imaging Spectrometer (MISR) on board the Earth Observing System (EOS) Terra (1999) satellite and a second identical MODIS sensor on the Aqua (2002) platform marked the beginning of a new era



in space-based aerosol remote sensing. AOT is routinely derived from MODIS observations by three distinct and independent algorithms: Dark Target algorithm (*Remer et al*., 2005; *Levy et al*., 2007; 2013), Deep Blue algorithm (*Hsu et al*., 2004, 2013), and the Multi-Angle Implementation of Atmospheric Correction (MAIAC) algorithm (*Lyapustin et al*., 2011, 2018).

In this paper, we investigate the applicability to ABI observations of existing algorithms to retrieve AOT over land from visible/near-IR measurements. The accuracy of the available multi-year long records of AOT products derived by the three MODIS algorithms is evaluated by direct comparison to ground-based observations from the Aerosol Robotic Network (AERONET) at multiple sites in TEMPO's area of regard. A brief description of MODIS aerosol

algorithms, their products, and satellite-ground collocation procedure are given in section 2. The results of the satellite-ground comparison of individual sites, composites of all sites, and error characterization are presented in section 3, followed by concluding remarks given in section 4.

## 2.  Datasets and Collocation Strategy

### 2.1 MODIS Dark Target Aerosol Product

The dark target (DT) algorithm of MODIS consists of two separate algorithms, a land component for the retrieval of aerosol properties over vegetated surfaces, and an over-ocean retrieval algorithm. The over-land DT algorithm exploits the top-of-atmosphere (TOA) reflectance measurements in three MODIS bands, i.e., 470 nm, 670 nm, and 2130 nm to simultaneously derive AOT at all three channels with an underlying assumption that the 2130 nm channel

contains information about coarse mode aerosol as well as the surface reflectance. The surface characterization is achieved through linear regression of surface reflectance in the 2130 nm and visible channels (470 nm, 670 nm) (*Kaufman et al*. 1997; *Remer et al*., 2005) accounting for the viewing geometry and "greenness" of land cover (*Levy et al*., 2007). DT attempts to perform retrieval on each 10 km grid box using a limited number of TOA reflectance observations after

discarding 50% brightest, 20% darkest, and cloudy pixels out of total 400 pixels at 500 meters resolution at nadir. DT is essentially a look-up table search algorithm which combines the pre-calculated spectral reflectance for a fine-mode and a coarse-mode dominated aerosol models with a proper weighting to represent the ambient aerosol properties over the target. The weighted-average spectral LUT reflectance values are compared against the TOA spectral



measurements of MODIS to find the best match in AOT. Each valid retrieval is assigned with an appropriate quality assurance confidence flag (QAC) with best retrievals are tagged with QAC=3. Over the land, the expected error for AOT (0.55 μm) with QAC=3 is estimated to be ±(0.05+15%), whereas that over the ocean is ±(0.03+5%) for retrievals with QAC≥1. A detailed

description of the DT Collection 6 algorithm is given in *Levy et al.* (2013) and also available online at URL https://darktarget.gsfc.nasa.gov/.

In addition to the 10-km AOT product, the MODIS DT algorithm also offers a higher resolution aerosol product at 3-km spatial scale. While both aerosol products closely resemble each other, the 3-km product differs from the original 10-km product in the manner in which the MODIS

pixels are ingested, organized, and selected by the aerosol algorithm (*Remer et al.*, 2013). The expected error associated with the 3-km aerosol retrievals over land is found to be greater than that of the 10-km product (*Remer et al.*, 2013), whereas over the ocean the errors are expected to be the same as the 10-km product. Over land, globally, the 3-km aerosol product is found to be 0.01 to 0.02 higher than the 10-km product, according to *Remer et al.*, (2013).

**2.2 MODIS Deep-blue Aerosol Product**

The MODIS deep-blue (DB) aerosol algorithm utilizes the radiance measurements at the blue wavelength (412 nm), where the surface reflectance over land is relatively lower than that at longer visible wavelengths, to retrieve the column AOT over vegetated as well as bright surfaces (*Hsu et al.*, 2004). The surface characterization scheme of DB adopts a hybrid approach that

applies the dynamical surface reflectance method for urban built-up and the precalculated surface reflectance database in conjunction with the normalized vegetation index in arid and semi-arid areas (*Hsu et al.*, 2013). The dynamical surface reflectance method allows greater spatial coverage of DB aerosol product by expanding the retrieval capability from the bright surfaces to all snow-free land surfaces, including vegetated areas. The cloudy pixels are screened by

examining the spatial variations of TOA reflectance at 412 nm, 1380 nm, and brightness temperatures in the 11 μm and 12 μm bands. DB performs retrievals on cloud-free and snow-free pixels at nominal 1x1 km spatial resolution, and then aggregates afterward to the 10x10 km retrieval box. Unlike the DT algorithm, DB provides prognostic uncertainty defined relative to DB-retrieved AOT rather than to AERONET AOT. The uncertainty estimates for the best quality

retrievals (QAC=3) is formalized as $\pm ([0.086+0.56\tau_{DB}] /[1/\mu_0+1/\mu])$, where $\tau_{DB}$ is AOT retrieved



by DB algorithm, $\mu_0$ and $\mu$ are the cosines of solar and view zenith angles for a given retrieval (*Sayer et al.*, 2013). A detailed description of the second generation, enhanced DB retrieval algorithm is given in *Hsu et al.*, (2013).

## 2.3 MODIS Multi-Angle Implementation of Atmospheric Correction Aerosol Product

The Multi-Angle Implementation of Atmospheric Correction (MAIAC) algorithm retrieves surface bi-directional reflection factor (BRF) and AOT by using the time series of MODIS measurements over both dark vegetated surfaces as well as bright targets (*Lyapustin et al.*, 2011). The surface characterization in MAIAC is carried out by deriving the spectral regression coefficients that relate the surface BRF in the blue (470 nm), green (550 nm), and shortwave infrared (2130 nm) bands of MODIS. MAIAC considers two discrete aerosol models, i.e., background and dust, similar to the ones adopted in MODIS dark target algorithm (*Levy et al.*, 2007). For identifying the smoke aerosols generated from biomass burning, MAIAC employs a "smoke test" to discriminate smoke from clouds (*Lyapustin et al.*, 2012). The smoke test relies on a relative increase of aerosol absorption at MODIS wavelength 412 nm as compared to 470–670 nm owing to multiple scattering and enhanced absorption by organic carbon released during biomass burning combustion. Each valid 1-km AOT retrievals of MAIAC is accompanied by the associated quality flags which describe the observed conditions. Since its introduction in 2011-2012, MAIAC algorithm has been continuously updated and evaluated regarding its accuracy and performance. For the latest Collection 061 release, the MAIAC AOD accuracy can be evaluated as ±0.05±0.15*AOD or even better (±0.05±0.1*AOD) as shown in a global validation analysis reported in *Lyapustin et al.* (2018). For a more detailed description of the MAIAC collection 6 algorithm, the reader is referred to *Lyapustin et al.* (2018).

## 2.4 Ground-based AERONET AOT Measurements

The Aerosol Robotic Network (AERONET) project is a ground-based federated network of globally distributed Cimel Sun photometers designed to do aerosol remote sensing (*Holben et al.*, 1998). Started in 1992, AERONET has expanded its network from a few sites in the early years to more than 500 sites across the globe currently. For more than 25 years, the project has provided long-term, continuous, and readily accessible public domain database of aerosol optical, microphysical, and radiative properties. AERONET data has been extensively used for



aerosol characterization and validation of satellite retrievals. Spectral AOTs from the direct Sun measurements are available nominally at 340, 380, 440, 500, 675, 870, and 1020 nm. In the present analysis, we employ AERONET Version 2, Level 2 (cloud-cleared and quality-assured) (*Holben et al.*, 2006) spectral AOT dataset from a total of 171 sites span across the United States

and Canada to evaluate the performance of three MODIS aerosol algorithms. Figure 1 displays the geographical distribution of AERONET sites with the corresponding temporal record (color-coded). Table 1 summarizes the datasets and their characteristics.

## 2.5 Satellite-ground Collocation Strategy

The three MODIS aerosol algorithms report AOT at different spatial resolutions. The DT

algorithm performs and reports AOT at 10 km and 3 km spatial resolution; DB performs retrievals at 1 km but aggregates afterward to the 10x10 km retrieval box, whereas the MAIAC algorithm retrieves and report AOT at a much higher resolution of 1 km spatial grid. While AOT from all three aerosol products corresponds to an area intercepted in their respective spatial grid cells representing the atmospheric conditions over a small region, the direct measurements of the

spectral AOT from AERONET sunphotometer are columnar point measurements. Furthermore, AERONET makes AOT measurements at an interval of 15 minutes, and the timings of Aqua/MODIS overpass may not closely match with that of AERONET measurements. Therefore, collocating both types of measurements requires a spatiotemporal window that can adequately match the spatially-averaged satellite AOT retrievals with the temporally-averaged

ground-based measurements. The spatiotemporal approach developed by *Ichoku et al.* (2002) has been adopted in several validation studies for validating MODIS aerosol products against the ground truth, such as from AERONET. The standard approach suggests comparing spatially averaged satellite retrievals in a 0.5° x 0.5° grid box centered at the ground site with the temporal averaged sunphotometer measurements of AOT within a time window of ±30 minutes of satellite

overpass time.

In this study, we introduce variations in the standard spatiotemporal window by applying changes in both spatial and temporal domains to assess the performance of MODIS aerosol products on different space-time scales. Four different spatiotemporal windows were formulated

that differ in the size of grid box centered at the AERONET site and corresponding time window around Aqua overpass time for averaging the AERONET AOTs. For the MAIAC and DB





products, the minimum number of 1-km satellite observations used by the respective algorithms in the aerosol retrieval is required to be set at 20% of the maximum possible 1-km pixels contained in the respective grid boxes. Since the DT algorithm discards 50% brightest and 20% darkest pixels out of total number of available 500-meter pixels in each 10 km and 3 km grid box

before performing the retrieval, the threshold for DT algorithm was set to 10%. The minimum number of AERONET Level 2 AOTs around the satellite overpass time is required to be at least two for all four variants of the collocation scheme. Table 2 lists the configurations of all four spatiotemporal windows designed for the satellite-ground collocation.

The wavelengths of AOT retrievals differ among the three MODIS aerosol algorithms. While the DT algorithm retrieves and reports AOT at 470, 660, and 2130 nm, DB retrievals are available at 412, 470, and 660 nm. MAIAC retrieves AOT at 470 nm and reports it at 550 nm. For a consistent comparison against AERONET, we choose the 470-nm as a reference wavelength at common to all three algorithms. AERONET Sunphotometer, on the other hand, does not directly

measures AOT at 470 nm but provides measurements at nearby wavelengths, i.e., 440 nm, 500 nm, and 670 nm. Using the Angstrom Exponent calculated from AOTs at these wavelengths, the AERONET AOT was estimated for the 470 nm wavelength following a linear regression on the AOT versus wavelength relation on a log-log space. The MODIS AOT retrievals at 470 nm were then directly compared against the interpolated AOTs of AERONET at the same wavelength. We

use the best quality AOT retrievals as identified in their respective quality assurance fields (i.e., QAC=2 and 3 for DT and DB) of all three aerosol products that are claimed to be higher in confidence and free of cloud contamination.

## 3. Results

### 3.1 MODIS versus AERONET AOTs: Individual Sites
Figure 2 shows scatter plots of MODIS versus AERONET AOT matchups for the selected individual sites located in Eastern NA. These sites are characterized by lower surface albedo during the spring and summer seasons due to increased green cover, and typically influenced by background and urban-industrial aerosols. Different color codes are used to display matchups

points derived following the different collocation approaches described in the previous section.



Each AOT dataset was co-located to AERONET independently. While the AOT retrievals from all three algorithms are generally well-correlated (R>0.90) with those of AERONET, the DT algorithm overestimates AOT (10-km product) with a positive bias (0.04-0.12) and relatively larger RMSE. On the other hand, MAIAC AOTs are found to be slightly under-estimated, albeit

with the lowest RMSE and the largest number of matchups among the three algorithms. The performance of DB algorithm is found be intermediate with relatively better statistics of the comparison than those of DT over sites *CCNY*, *Toronto*, and *Walker_Branch,* but inferior performance over sites *GSFC* and *Univ_Of_Huston*. Table 3 lists various statistical measures of MODIS-AERONET AOT matchups for a number of sites located in Eastern NA.

Figure 3 shows similar MODIS versus AERONET comparison, but for a subset of sites over the western NA characterized by bright surfaces and inhomogeneous surface elevation. The retrieved AOT by the three MODIS algorithms differs markedly over these sites. The DT algorithm, which is designed to produce accurate aerosol retrievals over dark surfaces, significantly overestimates AOTs particularly at a smaller spatial scale of the collocation domain. Noticeably, spatial

averaging of DT AOTs over a larger spatial scale (40x40 km$^2$) at the *Fresno* site provides significantly improved agreement with AERONET AOTs as reflected by the different measures of statistics included in the scatter plot. DB and AERONET AOT matchups over these sites are found to be less correlated but with reduced RMSE. Over the *Railroad_Valley* site, most AOTs matchups from DB under all four collocation approaches remained in the range 0.0-0.2, whereas

AERONET AOTs varied in the range 0.0-0.4. The MAIAC-AERONET comparison over these sites shows relatively better statistics than those of DT and DB comparisons with a significantly greater number of matchups, the higher correlation coefficient, and lower RMSE values. Various statistical measures of MODIS versus AERONET AOT matchups for selected western NA sites are listed in Table 3.

**3.2 MODIS versus AERONET AOTs: Composites for Eastern and Western North America**

This section describes the MODIS-AERONET comparison results obtained by accumulating matchups derived separately for all Eastern and Western NA sites. The top panel of Figure 4 shows the composite comparison of MODIS AOTs to those of AERONET for all Eastern NA

sites combined. The comparison includes matchups obtained following the collocation scheme



that averages satellite data in 40 x 40 km$^2$ spatial domain and AERONET data within ±30 minutes of Aqua overpass time. Satellite-ground matchup points are color-coded according to the density of data for each AOT bin of size 0.01 as depicted in the color bar. One of the striking features of the comparison is that the total number of MAIAC AOT data points collocated with

5      AERONET is significantly larger than those obtained from DB and DT (10-km and 3-km) comparisons. Quantitatively, MAIAC provides ~ 108% and 125% (83%) more matchups than DB and DT (3-km aerosol product), respectively. In addition to the higher frequency of AOT retrievals, MAIAC AOTs are found to compare better with those of AERONET with an overall lower RMSE (0.05) and a higher correlation (R=0.92). Conversely, the performance of DT 10-

10     km algorithm is relatively inferior in terms of the number of matchup points, larger RMSE and bias with the slope of satellite-ground relationship greater than unity. DB and MAIAC comparisons to AERONET provide slopes less than 1.0 mainly due to under-estimation (over-estimation) of retrievals at higher (lower) AOTs, but with overall improvement in the other statistical measures. Noticeably, the DT 3-km product owing to its higher spatial resolution

offered more matchups accompanied with increased correlation (~0.93) compared to the 10-km retrievals, albeit with much larger slope (1.38) and RMSE (0.09) values.

For the combined western NA sites comparison, MAIAC again provides a significantly larger number of matchup points, quantitatively ~144%, 220%, and 195% compared to DB 10-km, DT 10-km, and DT 3-km products, respectively, with relatively lowest RMSE (0.053) and the

highest correlation (0.84). However, the slope of the satellite-ground AOT relationship is found to be the lowest (0.754) with MAIAC results compared to those obtained from DB (0.835), DT 10-km (1.072) and DT 3-km (1.26) datasets. The intercepts of the relationships are found to be comparable though.

The results presented so far considered satellite-ground matchups obtained independently for

each MODIS aerosol product. Such comparison allows evaluation of both the relative accuracy of different products as well as the frequency of the retrievals, whereas the comparison imposed by the requirement of having AOT retrievals from all three algorithms simultaneously would provide only the relative accuracy assessment. Such comparison is shown in Figure 5 for eastern (top) and western (bottom) NA sites. Note that the number of matchups is identical for all three

algorithms and is drastically lower than the collocation points obtained when matched with



AERONET independently. Given the simultaneous measurements of AOT and equal sampling among the three algorithms, MAIAC provided relatively highest correlation (0.9 and 0.84) and lowest RMSE (0.053 and 0.052) over eastern and western NA sites, respectively. The slope of the satellite-ground relationship, however, was farthest from unity for MAIAC compared to
those of DT and DB results.

### 3.3 Impact of Surface Reflectance on AOT Retrievals

The surface characterization is a crucial step for delineating surface contribution from the TOA reflectance measurements to separate atmospheric signal for the aerosol retrieval. Earlier studies suggest that an absolute uncertainty of 0.01 in the estimation of surface reflectance in the visible
channels can produce an error of up to 0.1, i.e., approximately ten times, in the AOT retrieval from the satellites (*Kaufman et al.*, 1997; *Jethva et al.*, 2010). The three independent MODIS aerosol algorithms under consideration here employ different approaches to characterize the surface reflectance as briefly described in the data section. The DT algorithm estimates surface reflectance in the visible channels (470 and 660 nm) through a quasi-static regression between
the reflectance at 2130 nm and those of visible channels by accounting for the dependence of these relationships on scattering angle and NDVI. The surface characterization in the DB algorithm is achieved through a hybrid scheme that applies the dynamical surface reflectance method for urban built-up and the precalculated surface reflectance database in arid and semi-arid areas. The MAIAC algorithm, on the other hand, derives the spectral regression coefficients
dynamically that relate the surface reflectance in the 470 nm, 550 nm, and 2130 nm bands of MODIS.

In this section, we explore the relationship between the surface reflectance either assumed (DT and DB) or retrieved (MAIAC) and its impact of the accuracy of AOT retrieved from three algorithms. For this purpose, we consider MAIAC BRF retrievals (470 nm) as a working
reference dataset since it encompasses surface characterization over darker as well as brighter surfaces, offering a wide range of surface conditions, and also due to the fact that it is a retrieved quantity from the atmospheric correction procedure that dynamically captures the temporal variation of surface properties. The MAIAC BRF product at the time of conducting the present work hasn't been evaluated over North America region. However, some recent studies have
reported a significant increase in the accuracy of MAIAC surface reflectance compared to


MODIS standard products MOD09, MOD035 over tropical Amazon (*Hilker et al.* 2012; 2014; 2015; *Maeda et al.*, 2016). Furthermore, a study by *Chen et al.* (2017) found an improvement in the leaf area index (LAI) retrievals with the MODIS LAI/FPAR algorithm when using MAIAC instead of standard MODIS MOD09 input. Note that the sole purpose of using MAIAC surface

retrieval dataset here is to evaluate relative differences between satellite retrievals and ground measurements of AOD at varying surface brightness, which in no way constitutes a validation exercise of MAIAC surface retrievals over the study region nor it acts as a bias towards a particular algorithm.

Figure 6 shows the box and whisker plots of differences in the AOT (470 nm) between the collocated MODIS retrievals and AERONET measurements for eastern NA sites (top panel) and western NA sites (bottom panel). The collocated dataset of MODIS and AERONET within 40 km diameter centered at AERONET site and ±30 minutes of MODIS overpass was used in these calculations. The total number of samples obtained in each bin of surface BRF is depicted at the

top of each sub-plot. For the eastern NA sites, the mean and mode of error between the DT/DB retrievals and AERONET measurements show negligible dependence on MAIAC surface BRF with most matchups remained close to the no-error limit but with an increased spread in data at surface BRF>0.04. The error in MAIAC AOT retrievals, however, is found to be very small with the mean and mode for each bin close to no error throughout the entire range of BRF retrieved

over eastern NA. Also, the spread of error (10 to 90 percentile group) in the MAIAC-AERONET matchups is noted smaller with an error limit mostly confined to within ±0.1.

For the sites located in western NA, the error in DT-retrieved AOT (both 10-km and 3-km) exhibits a systematic behavior showing significant growth of error accompanied by the larger spread in the data population at relatively higher surface BRF (0.05-0.1). Also, note that no

sufficient matchups are found between DT and AERONET for conditions when MAIAC retrieved much higher values of surface BRF. The poor performance of the DT algorithm over brighter surfaces has been a known problem (*Levy et al.*, 2010), although it was expected that the DT collection 006 algorithm would yield a lower bias over bright surfaces (*Levy et al.*, 2013). The DT algorithm was primarily designed and developed for the aerosol retrieval over darker

vegetated surfaces, as the name suggests, and follows the principle that aerosols brighten the

scene, which over the brighter surfaces, breaks down. Moreover, aerosol loading over western NA is relatively low, resulting in an inferior signal from aerosols compared to that from a brighter background.

## 4. Concluding Remarks

In this paper, we have performed the accuracy assessment of three Aqua/MODIS products of aerosol optical depth derived from three independent algorithms using ground-based AERONET measurements over the North America region. This is, to our knowledge, the first attempt to simultaneously evaluate the relative performance of the three MODIS aerosol products, i.e., DT, BB, and MAIAC, over the region, which is in the field-of-view of currently operational GOES

geostationary platform and future TEMPO mission. A spatio-temporal collocation scheme of satellite retrievals with ground measurements was applied identically to all three satellite-based products, except for the relaxed required minimum number of retrievals for the DT algorithm which discards many sub-kilometer pixels prior to performing the aerosol inversion. The comparison was carried out over a number of AERONET sites situated mostly in the United

States, and a few in Canada for the period 2002-2016, and under two sets of configurations, 1) when retrievals from all three algorithms are available simultaneously, and 2) independent comparison against AERONET.

We find that the performance of all three aerosol algorithms is comparable over darker surfaces of eastern NA with the MAIAC algorithm providing marginally better results with the lowest

RMSE (0.05) and the highest correlation (0.90). For the same comparison, the DT and DB algorithms yield and RMSE of ~0.08 and correlation of 0.85 and 0.90, respectively. When assessed independently without having the requirement of simultaneous retrievals for all three algorithms, the resultant statistics of the MODIS-AERONET comparison remained almost similar. The most significant difference was the number of retrievals with MAIAC yielding

significantly more matchups with AERONET than the other two algorithms. MAIAC's number of available retrievals is more than double that of DT and slightly greater than twice that of DB.

Over the western NA, where the surface is characterized by steep changes in topography and brighter surface background, AOT retrievals from DT algorithm are found to be overestimated compared to that from AERONET with poorer RMSE, correlation, and bias of ~0.19, 0.59, and





0.10 respectively. In comparison, DB and MAIAC both showed a relatively robust match with AERONET with the resultant RMSE of ~0.05-0.06 and correlation of 0.80-0.84. Noticeably, the MAIAC dataset provided the maximum number of matchups (N=26277) compared to that of DB (N=10785) and DT (N=8207) – a factor of 2.44 and 3.20 higher that DT and DB matchups against AERONET, respectively.

The error in AOT characterized as a function of bi-directional surface reflectance retrieved from MAIAC reflects the ability of DB and MAIAC algorithms to retrieve AOT with practically no bias, whereas DT-retrieved AOTs are found to be systematically overestimated at higher values of surface reflectance (>0.05). The results reported here represent an objective, unbiased evaluation of the DT, DB, and MAIAC land AOT retrieval algorithms currently applied to MODIS observations. The detailed statistical evaluation of the performance of each of these three algorithms may be used as guidance in the development of inversion schemes to derive aerosol properties from ABI or other MODIS-like sensors. An accurate AOT product from GOES-ABI measurements would fulfill the GEO-CAPE stated need of an aerosol product that can be used for both climate and air quality applications.



## Acknowledgment

The authors acknowledge the support of the MODIS Adaptive Processing System (MODAPS) SIPS team (https://earthdata.nasa.gov/about/sips/sips-modaps) for processing and making the MODIS aerosol data available to the user community. The presented work was carried as an integral part of the GEO-CAPE Aerosol Working Group activities, and the authors thank group members for their valuable suggestions. Acknowledgments are due to the individual MODIS aerosol teams for their feedback on the correct interpretation and use of the data products evaluated in the present paper.



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



# List of Tables

**Table 1.** MODIS-AERONET aerosol datasets and their characteristics.

5 **Table 2.** Configurations of four spatiotemporal windows for the collocation of MODIS and AERONET AOT datasets. Acronyms: DT: Dark Target; DB: Deep Blue; MAIAC: Multi-Angle Implementation of Atmospheric Correction

**Table 3.** Statistical measures of MODIS-AERONET AOT (470 nm) matchups for sites in 10 eastern North America. Numbers in bold indicate relatively best performance in respective measures. Abbreviations: Lon.: Longitude, Lat.: Latitude, N: number of satellite-ground matchups, R: correlation, RMSE: root-mean-square-error between MODIS and AERONET, Bias: mean bias, Slope and Intercept: slope and intercept of the linear regression between MODIS and AERONET AOT matchups.

**Table 4**. Same as in Table 3 but for sites in western North America.





*Table 1 MODIS-AERONET aerosol datasets and their characteristics.*

| Dataset | Characteristics | | | |
|---|---|---|---|---|
| | Collection | Data | Product Resolution | In this study |
| **MODIS Dark Target 10-km Aerosol Product MYD04_L2** | 6.1 | Level 2 AOT at 470, 660, and 2100 nm | 10 km$^2$ at nadir | Use of only "good" (QAC=2) and "best" (QAC=3) quality retrievals |
| **MODIS Dark Target 3-km Aerosol Product MYD04_L2** | 6.1 | Level 2 AOT at 470, 660, and 2100 nm | 3 km$^2$ at nadir | Use of only "good" (QAC=2) and "best" (QAC=3) quality retrievals |
| **MODIS Deep Blue Aerosol Product Merged with MYD04_L2** | 6.1 | Level 2 AOT at 412, 470, and 660 nm | 10 km$^2$ at nadir | Use of only "good" (QAC=2) and "best" (QAC=3) quality retrievals |
| **MODIS MAIAC Aerosol Product MCD19A2** | 6.1 | Level 2 Daily L2G 1 km SIN Grid AOT at 470 and 550 nm | 1 km at nadir | Use of only "good" and "best" quality retrievals |
| **AERONET AOT Product** | Level 2.0 Version 2.0 | Spectral AOTs | Columnar point measurements | Cloud-cleared and quality assured data |



*Table 2 Configurations of four spatiotemporal windows for the collocation of MODIS and AERONET AOT datasets. Acronyms: DT: Dark Target; DB: Deep Blue; MAIAC: Multi-Angle Implementation of Atmospheric Correction*

| Spatial Grid km$^2$ | Required minimum number of satellite observations at 1 km | | | | ΔT = Time window between the satellite overpass and AERONET measurements | Minimum number of AERONET Level 2 observations within ΔT |
|---|---|---|---|---|---|---|
| | DT 10-km | DT 3-km | DB | MAIAC | | |
| **5** | 2 | 5 | 5 | 5 | 15 minutes | 2 |
| **10** | 10 | 20 | 20 | 20 | 15 minutes | 2 |
| **20** | 40 | 80 | 80 | 80 | 15 minutes | 2 |
| **40** | 160 | 320 | 320 | 320 | 30 minutes | 2 |





***Table 3 Statistical measures of MODIS-AERONET AOT (470 nm) matchups for sites in eastern North America. Numbers in bold indicate relatively best performance in respective measures.***

*Abbreviations: Lon.: Longitude, Lat.: Latitude, N: number of satellite-ground matchups, R: correlation, RMSE: root-mean-square-error between MODIS and AERONET, Bias: mean bias between the two datasets, Slope and Intercept: slope and intercept of the linear regression between MODIS and AERONET AOT matchups.*

| Sitename | Lon. | Lat. | N | R | RMSE | Bias | Slope | Intercept |
|---|---|---|---|---|---|---|---|---|
| | | | | | Dart Target/Deep Blue/MAIAC | | | |
| *Ames* | -93.78 | 42.02 | 311/342/431 | **0.85**/0.82/0.82 | 0.09/**0.07**/**0.07** | -0.04/**-0.02**/-0.03 | **0.98**/0.59/0.72 | -0.04/0.04/**0.02** |
| *Appalachian_State* | -81.69 | 36.22 | 228/212/**233** | **0.91**/0.84/0.84 | 0.05/0.05/**0.03** | **-0.01**/-0.03/**-0.01** | 1.33/0.59/**0.77** | -0.04/**0.00**/0.01 |
| *Billerica* | -71.27 | 42.53 | 299/285/373 | **0.95**/0.89/0.93 | 0.07/0.07/**0.05** | -0.05/0.02/**-0.01** | **1.13**/0.79/0.83 | -0.07/0.05/**0.01** |
| *BONDVILLE* | -88.37 | 40.05 | 505/693/789 | **0.91**/0.90/0.86 | 0.08/**0.05**/0.06 | -0.04/**0.00**/-0.01 | 1.17/**0.90**/0.87 | -0.07/**0.01**/0.01 |
| *Bratts_Lake* | -104.70 | 50.28 | 643/546/779 | 0.94/0.92/**0.95** | 0.15/0.12/**0.05** | 0.10/0.02/**0.00** | 1.40/1.34/**0.99** | 0.05/-0.03/**0.01** |
| *Brookhaven* | -72.89 | 40.87 | 141/40/237 | **0.98**/0.97/**0.98** | 0.08/0.06/**0.04** | 0.03/0.03/**-0.01** | 1.24/0.87/**0.94** | -0.02/0.05/**0.00** |
| *CARTEL* | -71.93 | 45.38 | 315/354/388 | 0.94/0.93/**0.96** | 0.06/**0.04**/**0.04** | **0.00**/**0.00**/-0.03 | 1.16/0.85/**0.91** | -0.03/0.02/**-0.01** |
| *Cart_Site* | -97.49 | 36.61 | 1073/1038/1410 | 0.80/0.81/**0.82** | 0.09/0.09/**0.05** | -0.07/-0.02/**0.00** | **0.91**/0.61/0.71 | -0.06/**0.02**/0.03 |
| *CCNY* | -73.95 | 40.82 | 331/461/688 | **0.93**/0.92/0.92 | 0.09/0.07/**0.06** | 0.03/0.03/**-0.02** | 1.15/**0.93**/0.76 | **0.00**/0.04/0.01 |
| *Chapais* | -74.98 | 49.82 | 127/209/263 | 0.96/0.96/**0.97** | 0.08/0.06/**0.04** | 0.03/**0.00**/-0.01 | 1.25/1.07/**0.99** | 0.00/**-0.01**/**-0.01** |
| *Dayton* | -84.11 | 39.78 | 217/235/265 | **0.91**/0.89/0.87 | 0.05/**0.04**/**0.04** | **0.00**/0.01/-0.02 | 1.20/0.81/**0.87** | -0.02/0.03/**-0.01** |
| *Easton_Airport* | -76.07 | 38.81 | 124/113/215 | **0.96**/0.91/0.91 | 0.08/0.07/**0.06** | **0.02**/0.04/-0.03 | 1.25/**0.90**/0.82 | -0.03/0.06/**0.00** |
| *Egbert* | -79.75 | 44.23 | 461/401/559 | **0.93**/0.89/0.92 | 0.06/0.06/**0.04** | **0.01**/0.04/**-0.01** | 1.27/0.92/**0.90** | -0.03/0.05/**0.00** |
| *Georgia_Tech* | -84.40 | 33.78 | 204/201/212 | **0.95**/0.88/0.94 | 0.07/**0.04**/**0.04** | -0.05/**0.01**/-0.02 | 1.31/0.80/**1.00** | -0.08/0.03/**-0.02** |
| *GSFC* | -76.84 | 38.99 | 1051/1084/1230 | **0.96**/0.90/0.94 | 0.06/0.07/**0.04** | **0.00**/0.03/-0.02 | 1.21/0.80/**0.88** | -0.03/0.06/**0.00** |
| *Halifax* | -63.59 | 44.64 | 94/147/542 | **0.94**/0.86/**0.94** | 0.06/0.06/**0.04** | 0.04/0.05/**0.00** | 1.30/0.91/**0.93** | **0.00**/0.06/0.01 |
| *Harvard_Forest* | -72.19 | 42.53 | 322/346/426 | **0.96**/0.90/0.95 | 0.06/0.06/**0.04** | **0.00**/**0.00**/-0.01 | 1.25/0.89/**0.92** | -0.03/0.01/**0.00** |
| *Howland* | -68.73 | 45.20 | 169/199/232 | 0.93/0.92/**0.95** | 0.07/0.07/**0.06** | -0.01/**0.00**/-0.02 | **1.02**/0.79/0.81 | -0.01/0.03/**0.00** |
| *Kellogg_LTER* | -85.37 | 42.41 | 145/154/182 | 0.95/0.94/**0.96** | 0.07/**0.05**/**0.05** | -0.02/**0.01**/-0.02 | 1.21/0.92/**0.98** | -0.05/**0.02**/**-0.02** |
| *KONZA_EDC* | -96.61 | 39.10 | 794/752/941 | 0.86/**0.90**/0.85 | 0.07/**0.04**/0.05 | -0.02/**0.01**/**-0.01** | **1.08**/0.85/0.77 | -0.03/**0.02**/**0.02** |
| *MD_Science_Center* | -76.62 | 39.28 | 582/641/841 | **0.95**/0.87/0.92 | 0.06/0.06/**0.05** | -0.02/**0.01**/-0.03 | 1.17/0.68/**0.77** | -0.05/0.06/**0.00** |
| *Pickle_Lake* | -90.22 | 51.45 | 164/345/413 | 0.92/0.90/**0.92** | 0.06/**0.04**/0.06 | 0.03/**-0.01**/**-0.01** | 1.25/0.89/**1.08** | **0.00**/**0.00**/-0.02 |
| *SERC* | -76.50 | 38.88 | 454/257/765 | **0.97**/0.95/0.96 | 0.07/0.05/**0.04** | **0.00**/0.03/-0.01 | 1.25/0.87/**0.94** | -0.04/0.05/**0.00** |
| *Sioux_Falls* | -96.63 | 43.74 | 602/606/765 | **0.92**/**0.92**/0.88 | 0.08/0.07/**0.06** | -0.03/-0.02/**-0.01** | 1.13/**1.11**/0.81 | -0.05/-0.04/**0.01** |
| *Thompson_Farm* | -70.95 | 43.11 | 421/388/559 | **0.94**/0.88/0.92 | 0.06/0.07/**0.05** | **-0.01**/0.02/-0.02 | **1.13**/0.82/0.82 | -0.03/0.05/**0.01** |
| *Toronto* | -79.47 | 43.97 | 447/421/559 | **0.94**/0.93/0.93 | 0.09/0.06/**0.05** | 0.04/**0.02**/-0.03 | 1.27/0.85/**0.89** | **-0.01**/0.04/**-0.01** |
| *UAHuntsville* | -86.65 | 34.73 | 140/133/154 | **0.95**/**0.95**/0.93 | 0.06/**0.04**/0.05 | -0.04/**-0.01**/-0.03 | 1.22/0.73/**0.88** | -0.07/0.02/**-0.01** |
| *UMBC* | -76.71 | 39.26 | 260/312/365 | **0.93**/0.79/0.89 | 0.06/0.06/**0.05** | -0.03/**0.02**/-0.02 | **1.180**/0.66/0.80 | -0.06/0.06/**0.01** |
| *Univ_of_Houston* | -95.34 | 29.72 | 421/420/593 | **0.92**/0.69/0.84 | **0.05**/0.10/**0.05** | 0.01/0.08/**-0.02** | **1.25**/0.70/0.75 | -0.02/0.11/**0.01** |
| *Walker_Branch* | -84.29 | 35.96 | 327/312/346 | **0.96**/**0.96**/0.95 | 0.07/**0.04**/**0.04** | **-0.01**/**-0.01**/-0.02 | 1.29/0.93/**0.95** | -0.06/**0.00**/-0.02 |
| *Wallops* | -75.48 | 37.94 | 359/204/610 | 0.94/0.94/**0.95** | 0.09/0.08/**0.06** | 0.05/0.05/**-0.02** | 1.08/**0.84**/**0.84** | 0.03/0.08/**0.01** |





***Table 4 Same as in Table 3 but for sites in western North America.***

| Sitename | Long. | Lat. | N | R | RMSE | Bias | Slope | Intercept |
|---|---|---|---|---|---|---|---|---|
| | | | | | Dark Target/Deep Blue/MAIAC | | | |
| *Bozeman* | -111.05 | 45.66 | 632/429/**695** | **0.97**/0.95/0.94 | **0.06**/0.07/**0.06** | 0.04/-0.03/**0.01** | **1.03**/0.92/0.76 | 0.03/-**0.02**/0.04 |
| *BSRN_BAO_Boulder* | -105.01 | 40.05 | 899/768/**1313** | **0.91**/0.82/0.86 | 0.08/0.07/**0.05** | 0.05/-0.04/**0.01** | 1.25/0.74/**0.86** | **0.02**/-**0.02**/0.03 |
| *CalTech* | -118.13 | 34.14 | 630/596/**748** | 0.71/0.53/**0.79** | 0.12/0.08/**0.06** | 0.08/-**0.01**/-0.03 | **1.03**/0.38/0.57 | 0.08/0.08/**0.03** |
| *El_Segundo* | -118.38 | 33.91 | 266/151/**715** | **0.55**/0.57/0.68 | 0.28/0.15/**0.05** | 0.23/0.12/**0.01** | 1.68/**0.84**/0.62 | 0.15/0.14/**0.05** |
| *Frenchman_Flat* | -115.94 | 36.81 | 138/707/**933** | 0.52/0.48/**0.67** | 0.27/0.06/**0.05** | 0.25/**0.01**/0.02 | 2.18/0.56/**0.73** | 0.18/**0.04**/0.04 |
| *Fresno_2* | -119.77 | 36.79 | 672/740/**773** | 0.80/0.82/**0.84** | 0.08/0.07/**0.06** | **0.01**/0.03/-0.02 | 1.07/0.88/0.69 | **0.00**/0.05/0.02 |
| *Fresno* | -119.77 | 36.78 | 1024/1076/**1100** | 0.74/**0.82**/0.79 | 0.08/**0.06**/0.07 | 0.01/**0.01**/-0.03 | **0.86**/0.72/0.58 | **0.03**/0.06/0.04 |
| *Goldstone* | -116.79 | 35.23 | 85/638/**1077** | 0.59/0.50/**0.68** | 0.23/**0.06**/**0.06** | 0.22/**0.02**/0.04 | 1.76/0.71/**0.76** | 0.18/**0.04**/0.06 |
| *Hermosillo* | -110.96 | 29.08 | 157/387/**451** | **0.83**/0.69/0.69 | **0.05**/0.07/**0.05** | 0.02/-0.05/-**0.01** | **1.14**/0.51/0.71 | **0.00**/**0.00**/0.03 |
| *HJAndrews* | -122.22 | 44.24 | 729/707/**767** | 0.91/0.88/**0.92** | 0.05/0.05/**0.03** | 0.02/-0.04/**0.01** | 1.15/0.65/**0.99** | **0.01**/-**0.01**/**0.01** |
| *Kelowna* | -119.37 | 49.96 | 287/221/**350** | **0.93**/0.85/**0.93** | 0.06/0.09/**0.04** | -0.01/-0.02/**0.00** | 1.10/1.14/**0.91** | -0.02/0.03/**0.01** |
| *Kelowna_UAS* | -119.40 | 49.94 | 326/253/**428** | **0.96**/0.88/0.91 | 0.07/0.07/**0.05** | -**0.01**/-0.03/-**0.01** | 1.09/**0.98**/0.88 | -0.02/-0.03/**0.01** |
| *Kirtland_AFB* | -106.51 | 34.95 | 131/214/**323** | 0.60/0.43/**0.77** | 0.08/0.05/**0.04** | 0.06/-0.03/**0.02** | 1.16/0.08/**0.90** | 0.05/**0.02**/0.03 |
| *La_Jolla* | -117.25 | 32.87 | 293/116/**815** | 0.71/0.68/**0.84** | 0.06/0.05/**0.04** | -0.01/0.01/**0.00** | **0.91**/0.52/0.74 | **0.00**/0.05/0.03 |
| *Maricopa* | -111.97 | 33.07 | 30/551/**672** | 0.78/0.52/**0.67** | 0.17/0.06/**0.05** | 0.15/-0.04/**0.02** | 1.84/0.43/**0.72** | 0.06/**0.02**/0.05 |
| *Missoula* | -114.08 | 46.92 | 745/626/**885** | **0.96**/0.83/**0.96** | **0.06**/0.15/**0.06** | -**0.01**/-0.05/-**0.01** | **1.06**/1.20/0.84 | -0.02/-0.08/**0.01** |
| *Monterey* | -121.86 | 36.59 | 757/446/**1053** | 0.88/0.72/0.85 | 0.08/0.07/**0.06** | -0.03/0.05/**0.01** | 1.24/0.81/**0.84** | -0.05/0.06/**0.03** |
| *NASA_Ames* | -122.06 | 37.42 | 78/71/**97** | 0.71/0.76/**0.85** | 0.06/0.06/**0.03** | **0.01**/0.05/**0.01** | **1.04**/0.73/0.81 | **0.00**/0.06/0.02 |
| *NEON-Boulder* | -105.27 | 40.01 | 58/45/**76** | 0.97/**0.98**/0.96 | 0.06/0.05/**0.04** | 0.03/-0.03/**0.01** | 1.17/0.82/**0.86** | **0.01**/-0.02/0.02 |
| *NEON_CVALLA* | -105.17 | 40.16 | 232/197/**313** | **0.94**/0.73/0.90 | 0.08/0.11/**0.05** | 0.03/-0.05/**0.01** | 1.21/**0.90**/0.83 | **0.00**/-0.03/0.02 |
| *Railroad_Valley* | -115.96 | 38.50 | 130/548/**1683** | 0.59/0.69/**0.74** | 0.25/0.06/**0.05** | 0.23/-**0.02**/0.03 | 1.69/0.26/**0.70** | 0.19/**0.02**/0.05 |
| *Red_Mountain_Pass* | -107.73 | 37.91 | 103/46/**168** | **0.79**/0.35/0.63 | 0.05/**0.04**/**0.04** | **0.04**/-**0.01**/0.03 | **1.05**/0.11/0.78 | 0.04/**0.03**/0.04 |
| *Rimrock* | -116.99 | 46.49 | 815/753/**1046** | 0.92/0.89/**0.92** | 0.17/0.14/**0.06** | 0.07/0.01/**0.03** | 1.94/1.76/**1.03** | -0.05/-0.08/**0.02** |
| *Rogers_Dry_Lake* | -117.89 | 34.93 | 24/326/**477** | 0.38/0.48/**0.63** | 0.16/0.09/**0.06** | 0.15/0.05/**0.03** | 1.31/**0.70**/0.57 | 0.14/0.07/**0.06** |
| *Sandia_NM_PSEL* | -106.54 | 35.06 | 184/225/**418** | 0.63/0.44/**0.73** | 0.11/**0.05**/0.06 | 0.07/-**0.03**/0.04 | 1.42/0.09/**0.96** | 0.04/**0.02**/0.04 |
| *Sevilleta* | -106.89 | 34.36 | 373/903/**1284** | 0.66/0.56/**0.76** | 0.16/0.06/**0.04** | 0.14/-0.04/**0.02** | 1.72/0.18/**0.82** | 0.09/**0.02**/0.03 |
| *TABLE_MOUNTAIN_CA* | -117.68 | 34.38 | 1093/1133/**1479** | 0.63/0.47/**0.68** | 0.14/0.06/**0.05** | 0.12/**0.04**/0.04 | 1.59/0.74/**0.89** | 0.09/**0.05**/0.05 |
| *Table_Mountain* | -105.24 | 40.13 | 333/295/**475** | **0.90**/0.89/0.84 | 0.06/0.06/**0.05** | 0.02/-0.04/**0.02** | **1.18**/0.54/0.79 | **0.00**/**0.00**/0.04 |
| *Trinidad_Head* | -124.15 | 41.05 | 292/138/**616** | 0.90/0.87/**0.95** | 0.08/0.07/**0.04** | 0.03/0.01/**0.01** | 1.31/1.27/**0.91** | **0.00**/-0.02/0.02 |
| *Tucson* | -110.95 | 32.23 | 274/407/**530** | 0.59/0.56/**0.81** | 0.19/**0.04**/0.05 | 0.17/-**0.01**/0.03 | 1.62/0.47/**0.83** | 0.13/**0.02**/0.04 |
| *UCLA* | -118.45 | 34.07 | 224/179/**275** | 0.67/0.53/**0.82** | 0.12/0.10/**0.07** | 0.06/**0.01**/-0.04 | **0.91**/0.44/0.58 | 0.07/0.10/**0.03** |
| *UCSB* | -119.85 | 34.42 | 840/481/**1062** | 0.79/0.71/**0.90** | 0.07/0.06/**0.05** | -0.05/-**0.02**/-0.02 | **0.76**/0.51/0.74 | -0.02/0.04/**0.01** |
| *Univ_of_Lethbridge* | -112.87 | 49.68 | 408/326/**546** | 0.92/0.91/**0.95** | 0.13/0.16/**0.05** | 0.09/**0.02**/0.02 | 1.36/1.67/**0.91** | 0.05/-0.05/**0.03** |
| *White_Sands_HELSTF* | -106.34 | 32.64 | 317/642/**1283** | **0.79**/0.59/0.73 | 0.17/**0.05**/0.06 | 0.16/-**0.01**/0.04 | 1.38/0.58/**0.86** | 0.13/**0.01**/0.05 |





# List of Figures

**Figure 1** a) Geographical distribution of AERONET sites over North America. Color codes represent the span of AERONET Level 2 data in years calculated from the total number of daily observations. b) An illustration of the spatiotemporal schemes for collocating the satellite retrievals with the ground measurements.

**Figure 2** Scatterplots comparing the aerosol optical depth (470 nm) retrieved from the three standard aerosol algorithms of MODIS against that of AERONET for selected sites over eastern, central, and southern N. A. Statistical measures of the comparison are depicted within each plot with different color codes denoting matchups obtained following the four spatiotemporal schemes, i.e., black, blue, green, and red for 5 km, 10 km, 20 km, and 40 km grid boxes.

**Figure 3** Same as in Figure 3 but for AERONET sites located in the western N. A.

**Figure 4** Scatterplots comparing MODIS-AERONET AOT matchups for all sites combinedly located in eastern N. A. (top panel) and western N. A. (bottom panel). MODIS-AERONET matchups derived independently without the requirement of having simultaneous measurements. The color codes denote the number density of matchups for each bin of AOT.

**Figure 5** Scatterplots comparing MODIS-AERONET AOT matchups for all sites combinedly located in eastern N. A. (top panel) and western N. A. (bottom panel). Only those satellite-ground matchups were included for which AOT retrievals/measurements from all four methods are available simultaneously. The color codes denote the number density of matchups for each bin of AOT.

**Figure 6** Difference in AOT (470 nm) between MODIS and AERONET as a function of coincident bi-directional reflectance retrievals from MAIAC aerosol algorithm for eastern N.A (a, top) and western NA (b, bottom). Data are represented as a box-and-whisker plot with the thick horizontal line as the median, black dot as mean, shaded boxes are covering 75 and 25 percentiles, and vertical lines as 1.5 times the interquartile range (25-75 percentile). The number of matchups for each bin is given at the top of the plot.





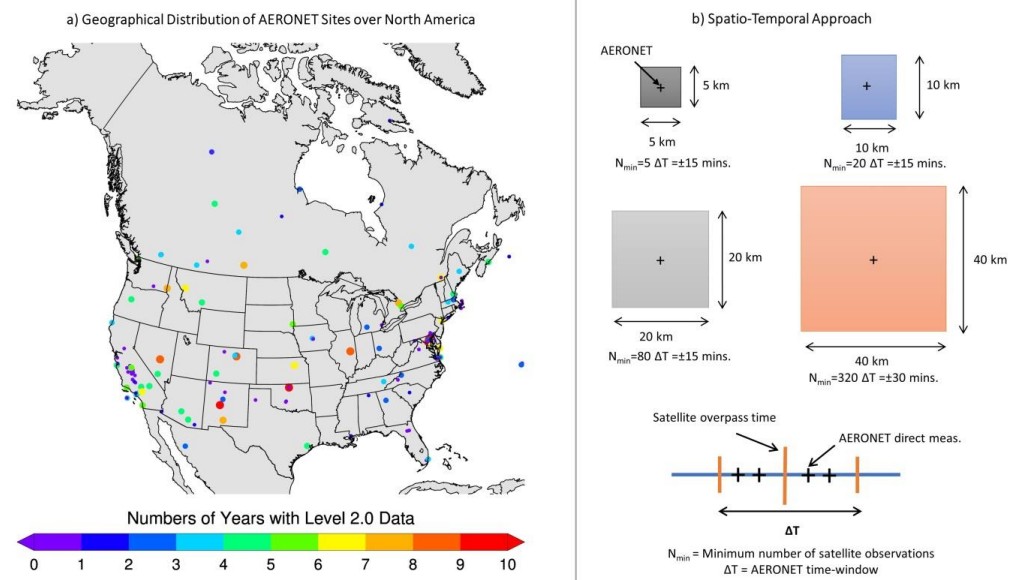

**Figure 1** a) Geographical distribution of AERONET sites over North America. Color codes represent the span of AERONET Level 2 data in years calculated from the total number of daily observations. b) An illustration of the spatiotemporal schemes for collocating the satellite retrievals with the ground measurements.




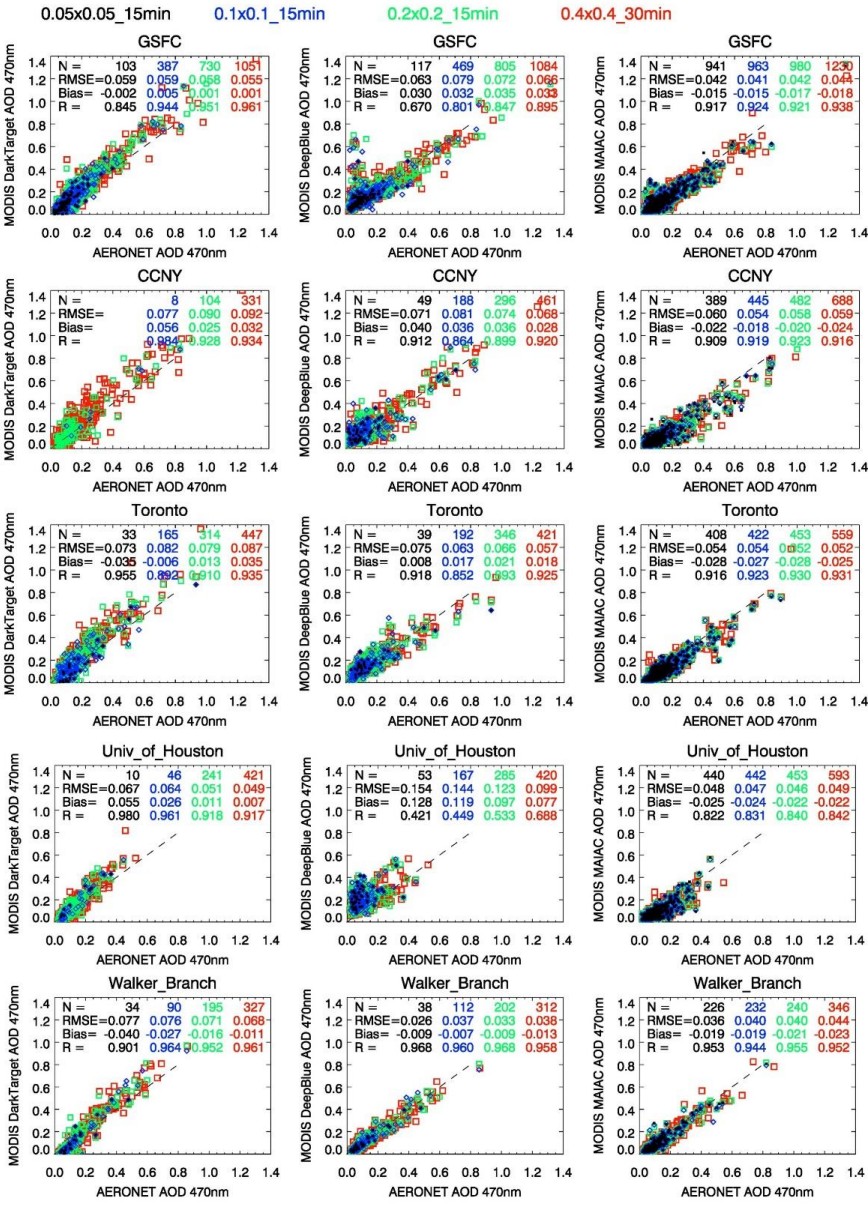

**Figure 2** Scatterplots comparing the aerosol optical depth (470 nm) retrieved from the three standard aerosol algorithms of MODIS against that of AERONET for selected sites over eastern, central, and southern N. A. Statistical measures of the comparison are depicted within each plot with different color codes denoting matchups obtained following the four spatiotemporal schemes, i.e., black, blue, green, and red for 5 km, 10 km, 20 km, and 40 km grid boxes.





**Figure 3** As in Figure 2 but for AERONET sites located in the western N. A.





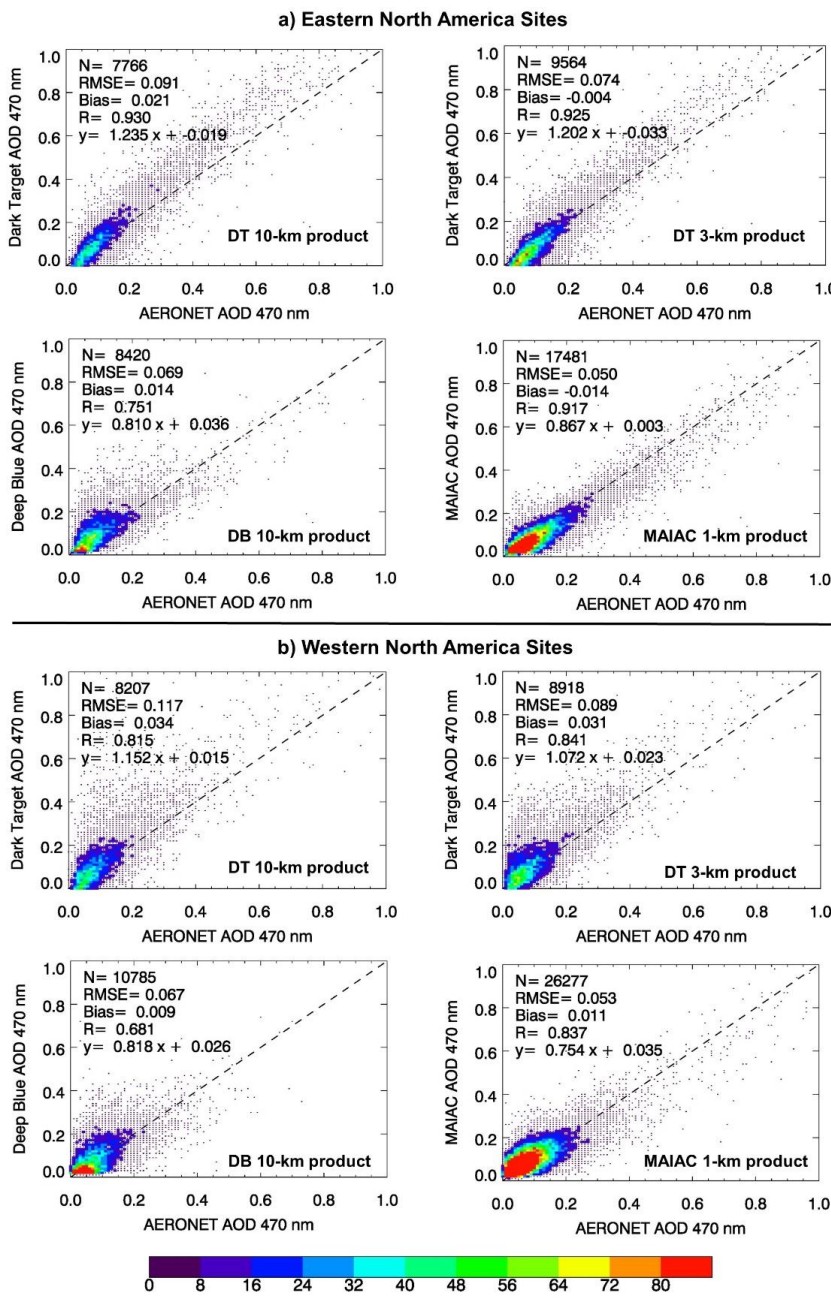

**Figure 4** Scatterplots comparing MODIS-AERONET AOT matchups for all sites combinedly located in eastern N. A. (top panel) and western N. A. (bottom panel). MODIS-AERONET matchups derived independently without the requirement of having simultaneous measurements. The color codes denote the number density of matchups for each bin of AOT.





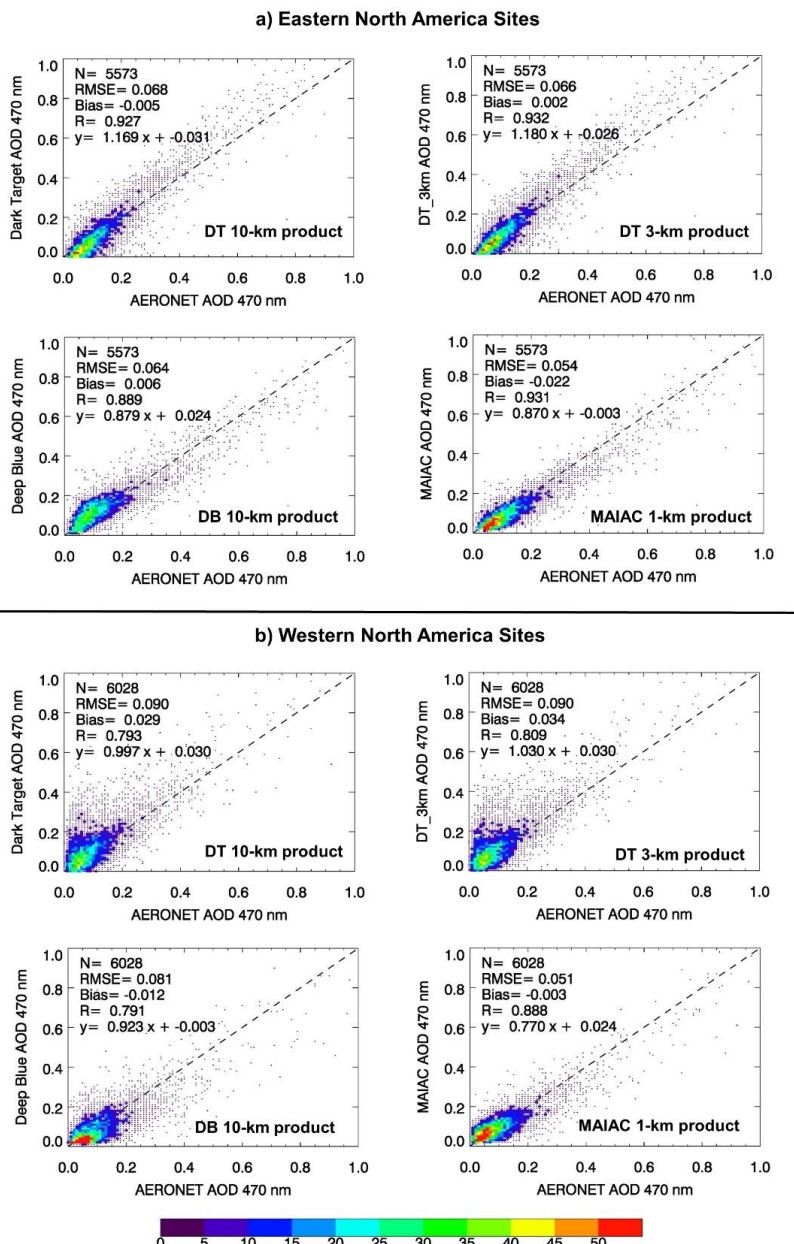

**Figure 5** Scatterplots comparing MODIS-AERONET AOT matchups for all sites combinedly located in eastern NA (top panel) and western NA (bottom panel). Only those satellite-ground matchups were included for which AOT retrievals/measurements from all four methods are available simultaneously. The color codes denote the number density of matchups for each bin of AOT.





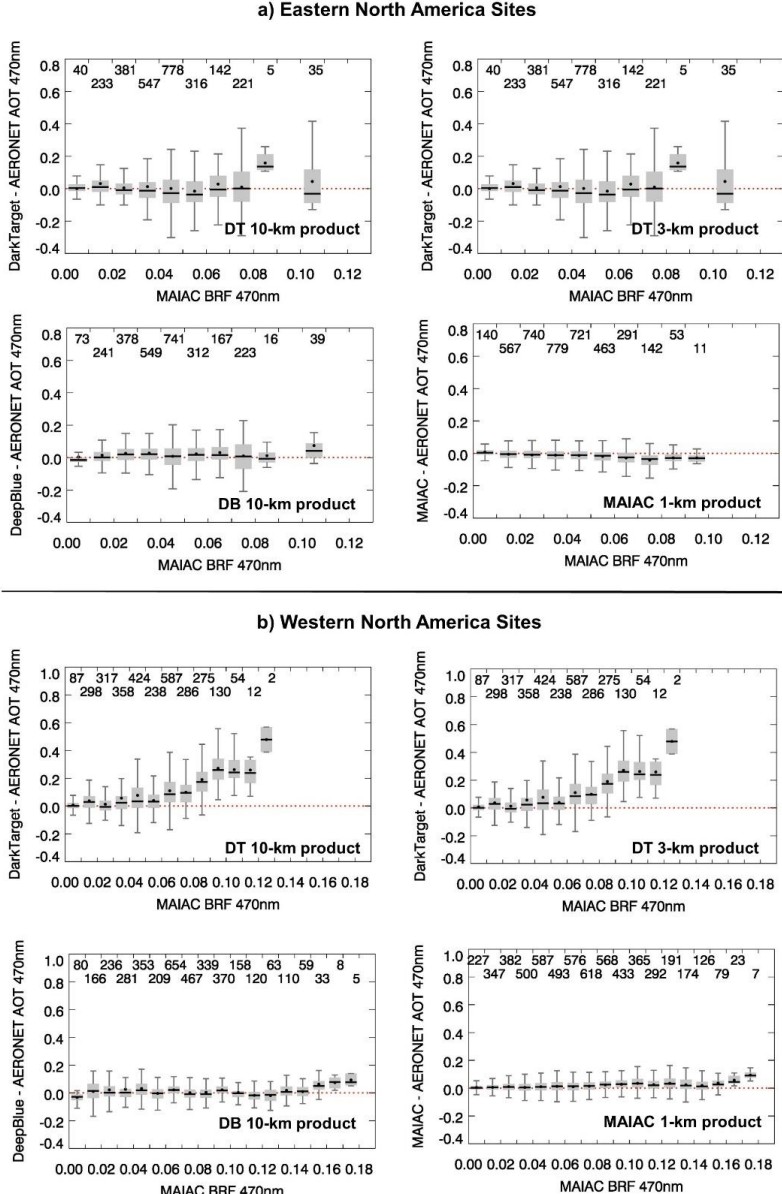

**Figure 6** Difference in AOT (470 nm) between MODIS and AERONET as a function of coincident bi-directional reflectance retrievals from MAIAC aerosol algorithm for eastern N.A (a, top) and western NA (b, bottom). Data are represented as a box-and-whisker plot with the thick horizontal line as the median, black dot as mean, shaded boxes are covering 75 and 25 percentiles, and vertical lines as 1.5 times the interquartile range (25-75 percentile). The number of matchups for each bin is given at the top of the plot.