# Peer review of "Accuracy Assessment of MODIS Land Aerosol Optical Thickness Algorithms using AERONET Measurements over North America"

_Atmospheric Measurement Techniques, 2019_

## Referee Comment (RC1) · Alexei Lyapustin (Referee) · 19 Apr 2019

This paper provides AERONET-based analysis of three aerosol products from DT, DB and MAIAC algorithms using an identical spatio-temporal collocation approach over North America. The study represents Eastern and Western USA separately. While the performance of the DB and DT algorithms has been thoroughly studied over the years, and global analysis of MAIAC has been recently reported in Lyapustin et al., 2018 (with growing number of regional analyses), the simultaneous performance comparison has been largely missing, except for that over the South Asia region published recently by Mhawish et al. The approach is sound, and results are clear. The language in some

places could be improved to achieve more clarity. My recommendation is to publish with minor revision, after authors address my comments below.

Throughout the paper: I suggest to replace GOES R/S with now standard GOES 16/17.

P. 3, Ln. 20: "The combination of sub-kilometer spatial resolution" It is 500m only in the Red band. Vis-NIR bands are at 1km, and 2.25um is at 2km resolution.

Ln. 28: Replace "Spectrometer" with Spectroradiometer for both MODIS and MISR

P.6, ln.20: Please, replace "061" with 6.0. Also, everywhere through the paper: MODIS DB and DT are Collection 6.1. MAIAC currently is Collection 6.0.

Ln.11: "MAIAC considers two discrete aerosol models": This is correct for a given location. However, MAIAC has 7 different regional aerosol models for different regions of the world. Besides, the DT algorithm tries to mix the background model with the dust model resulting in the fine mode fraction, whereas MAIAC uses either the background model or the dust model, if the dust has been detected.

Also, since the algorithm has rather significantly changed from 2011 to 2018 publications, I suggest that the initial reference adds the 2018 paper which really represents the MAIAC dataset used in this study.

P.6, ln26: "designed to do aerosol remote sensing". I suggest you remove this part as it doesn't sound right. It is ground-based sunphotometry, and "remote sensing" is usually associated with satellites.

Ln.30: What is "radiative" properties?

P.7, Ln.10: Given resolution of DB and DT is for the nadir only, it grows with the scan angle. "While AOT from all three aerosol products corresponds to an area intercepted in their respective spatial grid cells representing the atmospheric conditions over a small region, the direct measurements of the spectral AOT from AERONET sunphotometer are columnar point measurements." - It is not clear what you are saying, please

re-write.

Table 1 (MAIAC): Replace collection with 6.0, and remove "at nadir". MAIAC gives 1km2 everywhere.

Table 2: Add +- for the time interval. I don't understand the name "Spatial Grid km2" – is this the box size? The Figure 1 is very clear, but the name of the column, and also the description of the time-space collocation in the paper are very fuzzy. It will help if you improve the description.

Fig.5, Caption: remove "combinedly"

P.9, Ln.21: "relatively better statistics": This is a significantly better statistics. How do you define "relatively"?

P.11, Ln.2: The use of "relatively highest" is confusing: it is either highest or not. Also, MAIAC slope 0.87 over Eastern USA seems to be closer to 1 than the slope of DT (1.17) – doesn't it?

P. 11, Ln.22: The DB algorithm does not "assume" surface reflectance. The monthly surface reflectance database, binned over scattering angles, is derived from the previous years of measurements using the minimum reflectance method. In this sense, MAIAC approach is methodologically similar, though it derives SR spectral ratios via dynamical time series analysis from the latest measurements (on the fly).

P. 12: Just to note that Superczinsky et al. 2017 (JGR) found similar dependence on SR in comparison of VIIRS (a version of DT approach) and MAIAC.

Alexei.
* * *

---

## Referee Comment (RC2) · Anonymous Referee #2 · 23 Apr 2019

This paper consists of a unique analysis comparing three different algorithms (dark target, deep blue and MAIAC) applied to MODIS measurements over North America. This is the first inter-comparison of datasets produced with those three algorithms (except one earlier study over South Asia). The target area North America is of course relevant for many aspects impacting the environment and population of this continental area. The paper provides a detailed analysis using AERONET ground-based sun photometer measurements as reference and thus gives insight into the performance of the three algorithms, separately for Eastern and Western part of North America with known different environmental conditions and behavior of aerosol algorithms. I therefore recommend acceptance with minor revisions along my comments below.

[Figure]

1. Does the paper address relevant scientific questions within the scope of AMT? Users are always seeking guidance on choosing an appropriate dataset among several options offered by the retrieval community – the detailed analysis in this paper helps users understand the performance of three algorithms in direct comparison and choose along the criteria most relevant for their application. 2. Does the paper present novel concepts, ideas, tools, or data? This is clearly the case with for the first time comparing three mainly used MODIS algorithms over North America. 3. Are substantial conclusions reached? Yes, the paper gives clear conclusions for which criteria the algorithms perform similarly and where one of them shows better quality than the others. 4. Are the scientific methods and assumptions valid and clearly outlined? The validation exercises follow common practice and are described clearly. I appreciate that the analysis is conducted for all matchups and for common matchups which does best justice to all algorithms including their differences in coverage. In section 3.3 where a by-product from one of the three algorithms (surface BRF) is used to stratify the results of all three algorithms, this is explicitly explained. 5. Are the results sufficient to support the interpretations and conclusions? The conclusions are all supported by the statistical results in the tables and images. 6. Is the description of experiments and calculations sufficiently complete and precise to allow their reproduction by fellow scientists (traceability of results)? As validation follows common best practice, a reader in the field of aerosol retrievals should be able to reproduce the analysis with the descriptions provided. 7. Do the authors give proper credit to related work and clearly indicate their own new/original contribution? The new aspect of this paper is the target area (North America) in comparing three (not only the older two) MODIS algorithms. Underlying work of the developers of the three algorithms and the only other inter-comparison study (over South Asia) are cited. 8. Does the title clearly reflect the contents of the paper? Yes, but I would add the geographic area covered by this study. 9. Does the abstract provide a concise and complete summary? Yes, but I would rearrange it to put the future outlook at its end. 10. Is the overall presentation well structured and clear? The overall structure is appropriate and give good guidance to a reader. 11. Is

the language fluent and precise? As non-native English speaking, I am quite satisfied with the wording and have only few suggestions in my detailed comments. In some cases sentences tend to become very long – I would recommend to break those into two to ease reading. Also the tense used swaps sometimes from present to past. 12. Are mathematical formulae, symbols, abbreviations, and units correctly defined and used? There are mainly statistical results provided which are all using correct nomenclature. 13. Should any parts of the paper (text, formulae, figures, tables) be clarified, reduced, combined, or eliminated? The authors could think of moving tables 3 and 4 into an annex to shorten the main paper 14. Are the number and quality of references appropriate? Yes, all relevant material is provided. 15. Is the amount and quality of supplementary material appropriate? 16. N/A

General comments: I suggest to add "over North America" to the title to clearly identify the scope of the study already in the title. Please make sure to use consistent terminology: Through most of the paper you use "aerosol optical thickness", but in few places (conclusion, fig. 2) you use "aerosol optical depth" In the abstract and the introduction TEMPO / ABI as future perspective get too much weight and then in the paper it only appears again in the conclusion which may mislead a user on the scope of the paper. I therefore suggest to shorten this part (p. 1 / l. 11-16) in the abstract and put it at the end of the abstract (near p. 2 / l. 7/8) under future perspective. From the introduction I recommend to shift the part p. 3 /l. 3-24 to the conclusion, where it may get more attention.

Detailed comments: p. 1 / l. 17: delete "of" p. 1 / l. 28: add "allows FOR a" p. 2 / l. 3: write "show" instead of "showed" to remain consistent in tense with the first part of the sentence p.3 / l. 14: explain "PM" p.3 / l. 5: "we investigate the applicability to ABI observations" – I find this misleading since the paper neither analysis ABI datasets nor discusses relevant differences and similarities of MODIS and ABI in much detail. This is why I recommend to shift the discussion on the strategic potential into the conclusions. p. 5 /l. 1+2: delete "appropriate" (as vague wording); delete "are" p. 5 /l. 3+4: is should

be "over land" and "over ocean", without "the" p.5 / l.11: replace "greater" by "larger" p.5 / l.22: replace "greater" by "larger" p.5 / l. 24: delete "The" at the beginning of the sentence p. 6 / l.17: use singular: "Each valid . . . retrieval . . ." p. 7 /l. 2-5: this sentence is too long; you can delete " "to evaluate – aerosol algorithms" at its end and split the remaining sentence into two parts. p. 7 /l. 12: ". . . retrieves and reportS . . ." p. 7 / l. 12: delete "spatial grid" at the end p.7 / l. 22: delete "the" (ground truth) p. 8 /l. 4 it should read "out of A total" p. 8 / l. 13f: I would write "we choose 470 as reference wavelength common to all three . . ." p. 8 /l. 19ff: is there no q/a flag for MAIAC? If so this should be stated. p. 8 /l. 29: use singular "matchup" p. 9 / l. 4: I would write "under-estimated" – otherwise it may be miss-understood that MAIAC AOTs are under-estimated by AERONET. p. 9 / l. 1-3: I cannot find those numbers (0.04 – 0.12) in fig. 2 and also in tab. 3 the ranges of all stations are wider (MODIS DT bias -0.07 . . . 0.10) p. 9 /l. 18: use singular "AOT matchups" p. 9 /l. 22: replace "greater" by "larger"; delete "the" after the comma fig. 2 and fig. 3: the spatial matching criteria are indicated as =0.4x0.4 (misleading to degrees), while the text reads them as 40x40km2) – this inconsistency should be corrected p. 10 /l. 11: replace "greater" by "larger"; also two times "the" is missing in this sentence ("of the DT algorithm", "the satellite-ground" p. 10 / l. 21: replace "with MAIAC" by "for MAIAC" p. 11 / l. 2: use present time "provides" p. 11 / l. 10+11: delete "the" before "retrieval" and "satellites" p. 12 / l. 3: delete "the" before "leaf area" p. 12 / l. 4: add "the" before "MAIAC surface" p. 12 / l. 10: delete "the" before "box and" p. 12 / l. 17: correct to "remaining" p. 13 / l. 15-17: I suggest to swap 1 and 2 (same order as in the paper body) and to add "for each algorithm" to the "independent comparison against AERONET" p. 13 / l. 21: correct "an RMSE" instead of "and RMSE" p. 13 / l. 23: use "remain" instead of "remained" p. 13 / l. 23: "similar" is sufficient (i.e. delete "almost") p. 13 / l. 24: use "is" instead of "was" p. 13 / l. 26: replace "greater" by "larger" p. 13 / l. 26: add "the" before "DT algorithm" p. 14 / l. 1: use "show" instead of "showed" p. 14 / l. 3: use "provides" instead of "provided" p. 14 / l. 4: use "than" instead of "that"
* * *

---

## Referee Comment (RC3) · Anonymous Referee #3 · 24 Apr 2019

The paper validates the three existing MODIS aerosol products over North America. Several issues need to be clarified. Major comments: 1. The novelty of the manuscript has to be defined since there are so many similar works in the past decades. Although the author mentioned the paper as a preparation for the (TEMPO) mission, but again, so many similar works for the region, even in the conclusion part, the author mentioned the evaluation of three algorithm over NA is the "first time". 2. The paper used AERONET version 2 even the AERONET version 3 has been released for such a long time. Since the authors intend to understand three well-known aerosol products, the quality of those three products could be similar for most cases, thus an accurate reference is needed. Thus, I would like to ask the author to use version 3 for the validation.

3. The authors use the MAIAC reflectance product as reference" to understand the impact of surface to aerosol, this need to be further clarified since it is difficult to evaluate the "coupling" of MAIAC aerosol and surface product. I would ask the authors to use an independent surface product (MOD09 or MCD43).

Minor comments: P1 L12, full name of GOES R/S P1 L15 change "spectral coverage" to "wavelengths" P1 L16 change "currently used" to "existing" P1 L17 change "existing" to "three" P1 L17 change "of that derive" to "derived" P1 L20 full name of "Aqua-MODIS" P1 L20 change "carried out an independent evaluation of" to "evaluated" P1 L21 change "the retrieved AOT" to "the satellite retrieved AOT" P1 L24 are they really "consistently"? later you mentioned different criteria of pixel selection were used? P1 L25, delete "while" P1 L26 change "the MAIAC algorithm" to "and the MAIAC algorithm" P1 L28 change "finder" to "higher" P2 L1 is it really "error"? P2 L2-3, refer to major comment 3 P2 L4-9, these sentences are too general presented in abstract. P3 L6, what suspended particle means here? PM concentration? If so, how the vertical profile can be derived from ABI? P3 L14 – 16, this is not really accurate, the problem to get PM is to describe the vertical profile of aerosol and the humidity dependence of particle growth with respect to humidity. P3 L26 "over the globe" to "globally" P3 L27 "land and oceans" to "land and ocean" P4 L4, several sentence for the "similarities" and "differences" of those three algorithms have to be described. P4 L5, refer to major comment 1, a quick search online, we can already find similar work over other regions, if we focus over NA, there are much more publication for either two or single product(s) of them. Lyapustin, A., Wang, Y., Hsu, C., Torres, O., Leptoukh, G., Kalashnikova, O., Korkin, S., 2011b. Analysis of MAIAC dust aerosol retrievals from MODIS over North Africa. AAPP Phys. Math. Nat. Sci. 89. ELS XIII Conference, Vol. 89, Supplement No 1 Liu, N., Zou, B., Feng, H., Tang, Y., and Liang, Y.: Evaluation and comparison of MAIAC, DT and DB aerosol products over China, Atmos. Chem. Phys. Discuss., https://doi.org/10.5194/acp-2018-1339, in review, 2019. P4 L15-17, how DT separate land and ocean? And there is no description of ocean algorithm in this section. P4 L19-20, I suggest re-write this sentence, the assumption is the impact of fine mode

aerosol to 2.1 $\mu$m is ignorable P4 L25, how "cloudy pixels" detected? A reference is needed. P4 L26-28, aerosol type in DT is a location-time dependent prescribed type. P5 L1, how "best match" is found? P5L13, here the ocean algorithm suddenly appears. P5 L18, there is no" AOT over vegetated" in Hsu et al (2004) P5, L24 – 26, the dust screening should be mentioned P6, L3, "Hsu et al., (2013)" to "Hsu et al. (2013)" P6 L21, "$\pm 0.05 \pm 0.15$*AOD" to "$\pm(0.05+15\%)$", and harmonize AOT, AOD in the manuscript. P6 L21, "($\pm 0.05 \pm 0.1$*AOD)" to "$\pm(0.05+10\%)$" P7 L3, refer to major comment 2, why version 2? P8 L10 – 13, please check what the DT and DB retrieve? No AOT at 550 nm? P9 L3 -5, why? P9 L6, what is "better statistics" and why better? P9 L10 – 25, again, why? The authors need more explanations rather than simply list the statistics. Section 3.3, refer to major comment 3, I think the authors need to use an independent surface product. P13 L7-8, this statement is not enough as "novelty"
* * *

---

## Author Comment (AC1) · 26 Jun 2019

Dear Reviewer,

Thanks for offering your valuable comments on our manuscript # amt-2019-77. We have tried our best to incorporate all your suggestions, which have greatly improved the scientific merit of the paper. In the revision, two important and major changes have been applied according to the suggestions made by Reviewer # 3. These changes include,

1) use of the latest AERONET version 3 dataset (instead of version 2 used in the

original paper) 2) replacement of MAIAC BRF dataset with the MODIS standard BRF product (MOD09) in performing error characterization vs. BRF shown in Figure 6.

With these two changes, the entire analysis presented in the paper was reperformed to derive results tabulated in Table 3, 4, and Figure 1 through 6. While using AERONET version 3 dataset provided increased matchups and marginal change in the resultant statistics of the comparison (R, RMSE, bias, slope, intercept), the overall interpretation and conclusion of the MODIS-AERONET comparison for all three algorithms, i.e., DT, DB, and MAIAC, presented in the original paper haven't altered.

Following is the one-to-one response to each comment/suggestion made on the submitted manuscript.

RC: Referee's comment AR: Author's response

RC: Throughout the paper: I suggest to replace GOES R/S with now standard GOES 16/17. AR: Suggestion considered in the revision.

RC: P. 3, Ln. 20: "The combination of sub-kilometer spatial resolution" It is 500m only in the Red band. Vis-NIR bands are at 1km, and 2.25um is at 2km resolution. AR: The sentence is revised as, "The combination of 500 m to 2 km spatial resolutions and multispectral observations in the visible to shortwave-IR make the ABI an optimum sensor for the derivation of an aerosol optical thickness (AOT) product. . ."

RC: Ln. 28: Replace "Spectrometer" with Spectroradiometer for both MODIS and MISR AR: Corrected.

RC: P.6, ln.20: Please, replace "061" with 6.0. Also, everywhere through the paper: MODIS DB and DT are Collection 6.1. MAIAC currently is Collection 6.0. AR: Corrected.

RC: Ln.11: "MAIAC considers two discrete aerosol models": This is correct for a given location. However, MAIAC has 7 different regional aerosol models for different regions of the world. Besides, the DT algorithm tries to mix the background model with the dust

model resulting in the fine mode fraction, whereas MAIAC uses either the background model or the dust model, if the dust has been detected.

Also, since the algorithm has rather significantly changed from 2011 to 2018 publications, I suggest that the initial reference adds the 2018 paper which really represents the MAIAC dataset used in this study.

AR: The suggested information on MAIAC's choice of aerosol model and a reference of Lyapustin et al. (2018) are clarified in the revised paper.

"MAIAC considers two discrete aerosol models, i.e., background and dust for a given location, similar to the ones adopted in MODIS dark target algorithm (Levy et al., 2007). However, MAIAC prescribes 7 different regional aerosol models for different regions of the world and uses either background model or dust model, if the dust aerosols are detected."

"The MAIAC aerosol dataset used in the present study is derived using the latest Collection 6.0 version of the algorithm documented in Lyapustin et al. (2018), for which the AOT accuracy can be evaluated as ïĆś(0.05+15%)*AOT or even better ïĆś(0.05+10%)*AOT as shown in a global validation analysis."

RC: P.6, ln26: "designed to do aerosol remote sensing". I suggest you remove this part as it doesn't sound right. It is ground-based sunphotometry, and "remote sensing" is usually associated with satellites. AR: Both AERONET and satellite do remote sensing of aerosols albeit the former does it from ground, whereas the latter from space. To avoid the possible confusion, the sentence is modified as,

"The Aerosol Robotic Network (AERONET) project is a ground-based federated network of globally distributed Cimel Sun photometers designed to measure aerosol optical and microphysical properties (Holben et al., 1998)."

RC: Ln.30: What is "radiative" properties? AR: We meant "radiative" as the properties of aerosols largely determining the aerosols forcing on climate. Fundamentally, the

measures of aerosol such as AOT, SSA, and asymmetry parameter are the driving intrinsic properties that play a key role in modulating aerosol forcing. The sentence is now simplified by removing "radiative properties".

". . .and readily accessible public domain database of aerosol optical and microphysical properties."

RC: P.7, Ln.10: Given resolution of DB and DT is for the nadir only, it grows with the scan angle. "While AOT from all three aerosol products corresponds to an area intercepted in their respective spatial grid cells representing the atmospheric conditions over a small region, the direct measurements of the spectral AOT from AERONET sunphotometer are columnar point measurements." - It is not clear what you are saying, please re-write. AR: The sentence is revised as,

"While all three aerosol products report AOT at their respective nadir spatial resolutions, i.e., 10 km and 3 km for DT, 10 km for DB, and 1 km for MAIAC, representing the atmospheric conditions over the respective area intercepted at the ground,. . ."

RC: Table 1 (MAIAC): Replace collection with 6.0, and remove "at nadir". MAIAC gives 1km2 everywhere. AR: Corrected.

RC: Table 2: Add +- for the time interval. I don't understand the name "Spatial Grid km2" – is this the box size? The Figure 1 is very clear, but the name of the column, and also the description of the time-space collocation in the paper are very fuzzy. It will help if you improve the description. AR: +/- sign added to the time window column. "Spatial Grid km2" replaced with "Grid box size in km2". While we believe that the description of the collocation approach and spatiotemporal windows is adequate, we tried to improve the clarity by modifying the text in section 2.5

RC: Fig.5, Caption: remove "combinedly" AR: Figure 5 caption is revised as, "Scatter-plots comparing MODIS-AERONET AOT matchups obtained over all sites located in eastern NA (top panel) and western NA (bottom panel)."

RC: P.9, Ln.21: "relatively better statistics": This is a significantly better statistics. How do you define "relatively"? AR: We meant relatively w.r.t to the MODIS-AERONET statistical comparison obtained from other two aerosol algorithms.

RC: P.11, Ln.2: The use of "relatively highest" is confusing: it is either highest or not. Also, MAIAC slope 0.87 over Eastern USA seems to be closer to 1 than the slope of DT (1.17) – doesn't it? AR: The sentence is revised as, "Given the simultaneous measurements of AOT and equal sampling among the three algorithms, MAIAC provides highest correlation (0.9 and 0.84) and lowest RMSE (0.053 and 0.052) over eastern and western NA sites, respectively"

RC: P. 11, Ln.22: The DB algorithm does not "assume" surface reflectance. The monthly surface reflectance database, binned over scattering angles, is derived from the previous years of measurements using the minimum reflectance method. In this sense, MAIAC approach is methodologically similar, though it derives SR spectral ratios via dynamical time series analysis from the latest measurements (on the fly). AR: The surface reflectance (SR) dataset used in the DB algorithm is created from the full time-series and revised during each reprocessing. The SR dataset is essentially based on minimum reflectivity approach and binned by scattering angle, season, and NDVI with no time dimension except for the seasonal split. Over vegetated surfaces, DB follows the spectral ratio approach similar to that of DT. The hybrid method scales SR by regional BRDF shape, based on atmospheric correction near AERONET sites. We have further clarified this in the DB data section.

RC: P. 12: Just to note that Superczinsky et al. 2017 (JGR) found similar dependence on SR in comparison of VIIRS (a version of DT approach) and MAIAC. AR: The findings of Superczynski et al. (2017) supporting our results are mentioned in the revision as,

"Superczynski et al. (2017) further supports our findings using the AOT validation results of the Suomi-NPP Visible Infrared Imaging Radiometer Suite (VIIRS) aerosol algorithm essentially basing on the DT approach, where VIIRS-derived AOTs are found

to be bias significantly higher w.r.t to AERONET measurements over North America at larger values of coincident MAIAC-retrieved surface reflectance."

---

## Author Comment (AC2) · 26 Jun 2019

Dear Reviewer,

Thanks for offering your valuable comments on our manuscript # amt-2019-77. We have tried our best to incorporate all your suggestions, which have greatly improved the scientific merit of the paper. In the revision, two important and major changes have been applied according to the suggestions made by Reviewer # 3. These changes include,

1) use of the latest AERONET version 3 dataset (instead of version 2 used in the

original paper) 2) replacement of MAIAC BRF dataset with the MODIS standard BRF product (MOD09) in performing error characterization vs. BRF shown in Figure 6.

With these two changes, the entire analysis presented in the paper was reperformed to derive results tabulated in Table 3, 4, and Figure 1 through 6. While using AERONET version 3 dataset provided increased matchups and marginal change in the resultant statistics of the comparison (R, RMSE, bias, slope, intercept), the overall interpretation and conclusion of the MODIS-AERONET comparison for all three algorithms, i.e., DT, DB, and MAIAC, presented in the original paper haven't altered.

Following is the one-to-one response to each comment/suggestion made on the submitted manuscript.

RC: Referee's comment AR: Author's response

General comments:

RC: I suggest to add "over North America" to the title to clearly identify the scope of the study already in the title. AR: Following the suggestion, the title of the manuscript has been revised as,

"Accuracy Assessment of MODIS Land Aerosol Optical Thickness Algorithms using AERONET Measurements over North America"

RC: Please make sure to use consistent terminology: Through most of the paper you use "aerosol optical thickness", but in few places (conclusion, fig. 2) you use "aerosol optical depth" AR: we adopt aerosol optical thickness (AOT) terminology throughout the revised manuscript, i.e., in text as well in figures/legends.

RC: In the abstract and the introduction TEMPO / ABI as future perspective get too much weight and then in the paper it only appears again in the conclusion which may mislead a user on the scope of the paper. I therefore suggest to shorten this part (p. 1 / l. 11-16) in the abstract and put it at the end of the abstract (near p. 2 / l. 7/8) under future perspective. From the introduction I recommend to shift the part p. 3 /l. 3-24 to

the conclusion, where it may get more attention.

AR: The research work presented in the submitted paper was conducted as a part of the Geo-CAPE Aerosol Working Group at NASA Goddard in the context of evaluating existing aerosol algorithms for its possible application to the TEMPO/ABI synergy. The context has been adequately referred to in the abstract in order to highlight the objective of the paper, as well as in the Introduction (first two paragraphs) to begin with the motivation, and finally in the conclusion to close the loop.

Detailed comments:

RC: p. 1 / l. 17: delete "of" AR: Corrected as, "In this work, we evaluate three distinct aerosol algorithms of MODIS deriving aerosol optical thickness (AOT) over land surfaces using visible and near-IR observations."

RC: p. 1 / l. 28: add "allows FOR a" AR: Corrected as, "The higher spatial resolution of MAIAC product (1 km) allows a substantially larger number of matchups..."

RC: p. 2 / l. 3: write "show" instead of "showed" to remain consistent in tense with the first part of the sentence AR: Corrected.

RC: p.3 / l. 14: explain "PM" AR: PM is defined as "particulate matter"

RC: p.3 / l. 5: "we investigate the applicability to ABI observations" – I find this misleading since the paper neither analysis ABI datasets nor discusses relevant differences and similarities of MODIS and ABI in much detail. This is why I recommend to shift the discussion on the strategic potential into the conclusions. AR: The sentence is now revised as, "In this paper, we evaluate the accuracy of the available multi-year long records of AOT products derived by the three MODIS algorithms by a direct comparison to ground-based observations from the Aerosol Robotic Network (AERONET) at multiple sites in North America-an area or regard for both ABI and TEMPO field-of-views."

RC: p. 5 /l. 1+2: delete "appropriate" (as vague wording); delete "are" AR: The sen-
tence is revised as, "Each valid retrieval is assigned with a quality assurance confidence flag (QAC) with best retrievals tagged as QAC=3."

RC: p. 5 /l. 3+4: is should be "over land" and "over ocean", without "the" AR: Corrected.

RC: p.5 / l.11: replace "greater" by "larger" AR: The sentence is re-written as, "The expected error associated with the 3-km aerosol retrievals over land globally is found to be 0.01 to 0.02 higher than that of 10-km product (Remer et al., 2013)."

RC: p.5 / l.22: replace "greater" by "larger" AR: Changes accepted.

RC: p.5 / l. 24: delete "The" at the beginning of the sentence AR: Corrected.

RC: p. 6 / l.17: use singular: "Each valid : : : retrieval : : :" AR: Corrected.

RC: p. 7 /l. 2-5: this sentence is too long; you can delete " "to evaluate – aerosol algorithms" at its end and split the remaining sentence into two parts. AR: Changes accepted.

RC: p. 7 /l. 12: ": : : retrieves and reports. . ." AR: Corrected.

RC: p. 7 / l. 12: delete "spatial grid" at the end AR: Changes accepted.

RC: p.7 / l. 22: delete "the" (ground truth) AR: Corrected. RC: p. 8 /l. 4 it should read "out of A total" AR: Corrected.

RC: p. 8 / l. 13: I would write "we choose 470 as reference wavelength common to all three. . ." AR: The sentence is re-written as, "..we choose 470 nm as a reference wavelength due to the fact that all three algorithms actually retrieve AOT at this common wavelength."

RC: p. 8 /l. 19: is there no q/a flag for MAIAC? If so this should be stated. AR: MAIAC aerosol product does provide quality flags for each 1-km pixel retrieval, which is mentioned in section 2.3. We use best quality retrieval pixels which are free of cloud contamination.

RC: p. 8 /l. 29: use singular "matchup" AR: Corrected.

RC: p. 9 / l. 4: I would write "under-estimated" – otherwise it may be miss-understood that MAIAC AOTs are under-estimated by AERONET AR: Corrected.

RC: p. 9 / l. 1-3: I cannot find those numbers (0.04 – 0.12) in fig. 2 and also in tab. 3 the ranges of all stations are wider (MODIS DT bias -0.07 . . .0.10). AR: Since the statistical comparison results printed in Figure 2 are site dependent, the sentence has been simplified without mentioning the numbers as follows,

"While the AOT retrievals from all three algorithms are generally well-correlated (R>0.90) with those of AERONET, MAIAC AOTs are found to be slightly under-estimated, albeit with the lowest RMSE and the largest number of matchups among the three algorithms."

RC: p. 9 /l. 18: use singular "AOT matchups" AR: Corrected.

RC: p. 9 /l. 22: replace "greater" by "larger"; delete "the" after the comma fig. 2 and fig. 3: the spatial matching criteria are indicated as =0.4x0.4 (misleading to degrees), while the text reads them as 40x40km2) – this inconsistency should be corrected AR: "greater" is replaced with "larger" The spatial matching criteria are mentioned in km2 consistently throughout the manuscript including figures.

RC: p. 10 /l. 11: replace "greater" by "larger"; also two times "the" is missing in this sentence ("of the DT algorithm", "the satellite-ground" AR: Corrected.

RC: p. 10 / l. 21: replace "with MAIAC" by "for MAIAC" AR: Corrected.

RC: p. 11 / l. 2: use present time "provides" AR: Corrected.

RC: p. 11 / l. 10+11: delete "the" before "retrieval" and "satellites" AR: Corrected.

RC: p. 12 / l. 3: delete "the" before "leaf area" AR: Corrected.

RC: p. 12 / l. 4: add "the" before "MAIAC surface" AR: Corrected.

RC: p. 12 / l. 10: delete "the" before "box and" AR: Corrected.

RC: p. 12 / l. 17: correct to "remaining" AR: Corrected.

RC: p. 13 / l. 15-17: I suggest to swap 1 and 2 (same order as in the paper body) and to add "for each algorithm" to the "independent comparison against AERONET" AR: Suggestion accepted and included in the revision.

RC: p. 13 / l. 21: correct "an RMSE" instead of "and RMSE" AR: Corrected.

RC: p. 13 / l. 23: use "remain" instead of "remained" AR: Corrected.

RC: p. 13 / l. 23: "similar" is sufficient (i.e. delete "almost") AR: Corrected.

RC: p. 13 / l. 24: use "is" instead of "was" AR: Corrected.

RC: p. 13 / l. 26: replace "greater" by "larger" AR: Corrected.

RC: p. 13 / l. 26: add "the" before "DT algorithm" AR: Corrected.

RC: p. 14 / l. 1: use "show" instead of "showed" AR: Corrected.

RC: p. 14 / l. 3: use "provides" instead of "provided" AR: Corrected.

RC: p. 14 / l. 4: use "than" instead of "that" AR: Corrected.
* * *

---

## Author Comment (AC3) · 26 Jun 2019

Dear Reviewer,

Thanks for offering your valuable comments on our manuscript # amt-2019-77. We have tried our best to incorporate all your suggestions, which have greatly improved the scientific merit of the paper. In the revision, two important and major changes have been applied according to the suggestions made by Reviewer # 3. These changes include,

1) use of the latest AERONET version 3 dataset (instead of version 2 used in the

original paper) 2) replacement of MAIAC BRF dataset with the MODIS standard BRF product (MOD09) in performing error characterization vs. BRF shown in Figure 6.

With these two changes, the entire analysis presented in the paper was reperformed to derive results tabulated in Table 3, 4, and Figure 1 through 6. While using AERONET version 3 dataset provided increased matchups and marginal change in the resultant statistics of the comparison (R, RMSE, bias, slope, intercept), the overall interpretation and conclusion of the MODIS-AERONET comparison for all three algorithms, i.e., DT, DB, and MAIAC, presented in the original paper haven't altered.

Following is the one-to-one response to each comment/suggestion made on the submitted manuscript.

RC: Referee's comment AR: Author's response

Minor comments:

RC: P1 L12: full name of GOES R/S AR: The full name of GOES, Geostationary Operational Environmental Satellites, is referred in the abstract

RC: P1 L15: change "spectral coverage" to "wavelengths" AR: The suggestion is considered.

RC: P1 L16: change "currently used" to "existing" AR: The suggestion is considered.

RC: P1 L17: change "existing" to "three" and change "that of derive" to "derived" AR: The sentence has been revised according to the suggestion.

RC: P1 L20: full name of "Aqua-MODIS" AR: MODIS is defined earlier in the abstract.

RC: P1 L20: change "carried out an independent evaluation of" to "evaluated" AR: Suggestion considered in the revision.

RC: P1 L21: change "the retrieved AOT" to "satellite retrieved AOT" AR: Changed.

RC: P1 L24: are they really "consistently"? later you mentioned different criteria of

pixel selection were used? AR: We meant that the collocation procedure was applied to all three algorithms as identically as possible. Table 2 lists the configuration adopted for both satellite and ground datasets. For AERONET, the required minimum number of AOT measurements was set to 2 irrespective of the size of spatio-temporal window. The minimum number of satellite observations is selected depending on the size of spatial window. For DT-3km, DB, and MAIAC, the thresholds in min. number are set identical, whereas for the DT-10km product, the thresholds are relaxed to half to allow more matchups since a dark target algorithm uses limited number of 500m pixels in the retrieval after discarding 20% darkest and 50% brightest pixels in 10 km grid box.

RC: P1 L25: remove "while" and P1 L26: change the "MAIAC algorithm" to "and the MAIAC algorithm" AR: The sentence is now restricted according the suggestion.

RC: P1 L28: change "finer" to "higher" AR: The sentence is now rewritten as "The higher spatial resolution of MAIAC product (1 km) allows..."

RC: P2 L1: is it really "error" AR: The AERONET AOT due to its high accuracy (∼0.01) is considered as ground-truth, and therefore, the difference between satellite and ground AOTs is treated as error in the satellite retrievals.

RC: P2 L2-3: refer to major comment

RC: P2 L2-3: these sentences are too general presented in abstract. AR: Here, we close the abstract by emphasizing the usefulness of derived results, which may provide a guidance in the development of the aerosol algorithms for the aerosol retrievals from ABI or other MODIS-like sensors.

RC: P3 L6: what suspended particle means here? PM concentration? If so, how the vertical profile can be derived from ABI? AR: One of the goals of TEMPO-GOES synergy is to retrieve the mean aerosol layer height and single-scattering albedo using information in the near-UV from TEMPO by constraining the observed AOD (interpolated to near-UV) from ABI.

RC: P3 L14 – 16, this is not really accurate, the problem to get PM is to describe the vertical profile of aerosol and the humidity dependence of particle growth with respect to humidity. AR: We concur with the understanding here that neither TEMPO nor ABI can alone provide detailed vertical profiles of aerosols. However, the synergy between the two sensors can offer the mean aerosol layer height retrieved using information from the near-UV wavelengths with a constrain of AOT obtained from ABI using visible channels. The combined information of columnar AOT and aerosol layer height, therefore, help estimate the PM load when used as an input to the computation scheme equipped with other assumptions of meteorological variables including relative humidity. The sentences following to this claim in L14-L16 clearly states that the role of synergy between the two sensors.

RC: P3 L26 "over the globe" to "globally" AR: Changed.

RC:P3 L27 "land and oceans" to "land and ocean" AR: Changed.

RC: P4 L4, several sentence for the "similarities" and "differences" of those three algorithms have to be described. AR: Since detailed description of each algorithm (DT, DB, MAIAC) is given in papers published by the respective groups, we exercise brevity here and refer the readers to these paper for accessing details of each algorithm. However, we have tried to describe the major components of each algorithm, i.e., aerosol model and surface characterization, in the paper.

RC: P4 L5, refer to major comment 1, a quick search online, we can already find similar work over other regions, if we focus over NA, there are much more publication for either two or single product(s) of them.

Lyapustin, A., Wang, Y., Hsu, C., Torres, O., Leptoukh, G., Kalashnikova, O., Korkin, S., 2011b. Analysis of MAIAC dust aerosol retrievals from MODIS over North Africa. AAPP Phys. Math. Nat. Sci. 89. ELS XIII Conference, Vol. 89, Supplement No 1

Liu, N., Zou, B., Feng, H., Tang, Y., and Liang, Y.: Evaluation and comparison of

MAIAC, DT and DB aerosol products over China, Atmos. Chem. Phys. Discuss., https://doi.org/10.5194/acp-2018-1339, in review, 2019. P4 L15-1.

how DT separate land and ocean? And there is no description of ocean algorithm in this section.

AR: We concur with the reviewer that several papers before ours have validated MODIS DT and DB aerosol products, either together or alone, over different parts of the world, including North America. However, to our knowledge, our paper is the first attempt comparing all three existing aerosol products (DT, BD, MAIAC) simultaneously following a near-identical collocation approach against AERONET over North America region.

Since the main objective of the paper was to validate satellite retrievals of AOD over land, no emphasis was given to the over-ocean algorithm and its description in the manuscript.

RC:P4 L19-20, I suggest re-write this sentence, the assumption is the impact of fine mode aerosol to 2.1 ïA∎m is ignorable AR: The sentence has been re-written as,

"The over-land DT algorithm exploits the top-of-atmosphere (TOA) reflectance measurements in three MODIS bands, i.e., 470 nm, 670 nm, and 2130 nm to simultaneously derive AOT at all three channels with an underlying assumption that the impact of fine mode aerosol to 2130 nm signal is ignorable, and that the 2130 nm channel contains information about coarse mode aerosol as well as the surface reflectance"

RC: P4 L25, how "cloudy pixels" detected? A reference is needed. AR: A sentence mentioning the references and primary method to screen the cloudy pixels is added here. Since these references and ATBD describe cloud masking adequately, we don't include its details in this paper.

"The DT over-land algorithm screens cloudy pixels following a series of tests that rely on using absolute magnitude and spatial variability at 470 nm (500 m resolution) and 1380 nm (1 km resolution), the details of which are given in Martins et al., (2002) and

[Figure]

Levy et al., (2013)."

RC: P4 L26-28, aerosol type in DT is a location-time dependent prescribed type. AR: The sentence is re-written to reflect location-time dependent aerosol type feature of DT algorithm as,

"DT is essentially a look-up table search algorithm which combines the pre-calculated spectral reflectance of the location-time dependent aerosol models comprised of dominant fine and coarse modes with a proper weighting to represent the ambient aerosol properties over the target."

RC: P5 L1, how "best match" is found? AR: The sentence is re-written as,

"The weighted-average spectral LUT reflectance values are compared against the TOA spectral measurements of MODIS to find the best match in AOT yielding least square difference between simulated and observed reflectances."

RC: P5L13, here the ocean algorithm suddenly appears AR: The sentences have been simplified as,

"The expected error associated with the 3-km aerosol retrievals over land globally is found to be 0.01 to 0.02 higher than that of 10-km product (Remer et al., 2013)."

RC: P5 L18, there is no" AOT over vegetated" in Hsu et al (2004) AR: The sentence is now corrected as,

". . .where the surface reflectance over land is relatively lower than that at longer visible wavelengths, to retrieve the column AOT over bright surfaces (Hsu et al., 2004) as well as vegetated areas (Hsu et al., 2013)."

RC: P5, L24 – 26, the dust screening should be mentioned AR: Additional information on dust screening is added as,

"The enhanced second generation of DB algorithm identifies mineral dust aerosols based on the brightness temperature difference between infrared channels 8.6 ïA■m

and 11 $\mu$m as dust often produces stronger absorption at 8.6 ï︢A■m than that at 11 $\mu$m providing a robust way to detect strongly absorbing dust such as the silicates (Hsu et al., 2013)."

RC: P6, L3, "Hsu et al., (2013)" to "Hsu et al. (2013)" AR: Corrected.

RC: P6 L21, "ï︢Ćś0.05ï︢Ćś0.15*AOD" to "ï︢Ćś (0.05+15%)", and harmonize AOT, AOD in the manuscript. AR: Corrected. Also, we adopt aerosol optical thickness (AOT) terminology throughout the manuscript.

RC: P7 L3, refer to major comment 2, why version 2? AR: At the time of performing the present analysis (2016-2018), AERONET version 3 dataset wasn't published to the general public, and therefore not used. However, since now a complete version 3 data is available for use, we have re-performed the entire validation analysis using the latest AERONET v3 dataset.

RC: P8 L10 – 13, please check what the DT and DB retrieve? No AOT at 550 nm? AR: None of the three algorithms retrieve AOT at 550 nm. The DT algorithm retrieves and reports AOT at 470, 660, and 2130 nm, DB retrievals are available at 412, 470, and 660 nm, and MAIAC retrieves AOT at 470 nm and reports it at 550 nm.

RC: P9 L3 -5, why? AR: We believe that the overestimation in AOT shown by DT algorithm could primarily be due to the following few reasons: 1) inadequate characterization of surface reflectance, 2) choice of aerosol model, 3) non-optimum selection of 500 m resolution pixels, and 4) some minimal cloud contamination in the aerosol pixels (although it can't explain the totality of the overestimation). The main objective of the paper is to report the validation results instead of diagnosing the errors in detail. It is up to the algorithm development teams to analyze these results and figure out the probable causes of errors.

RC: P9 L6, what is "better statistics" and why better? P9 L10 – 25, again, why? AR: We meant here that the measures of comparison, i.e., RMSE, bias, and slope-intercept of

satellite vs. ground AOTs, are relatively better or improved. For instance, lower RMSE, bias, and intercept against little higher ones.

RC: P9 L10 – 25, again, why? The authors need more explanations rather than simply list the statistics. AR: Regarding why better comparison, please refer to the response given just above.

RC: Section 3.3, refer to major comment 3, I think the authors need to use an independent surface product. AR: We have replaced MAIAC surface BRF database with MODIS standard MOD09 product to reproduce revised Figure 6 showing error characterization as a function of surface BRF. The original Figure 6 using MAIAC BRF is now placed under supplementary material.